# OATS: ONLINE DATA AUGMENTATION FOR TIME SERIES FOUNDATION MODELS

## ABSTRACT

Time Series Foundation Models (TSFMs) are a powerful paradigm for time series analysis and are often enhanced by synthetic data augmentation to improve the training data quality. Existing augmentation methods, however, typically rely on heuristics and static paradigms. Motivated by dynamic data optimization, which shows that the contribution of samples varies across training stages, we propose OATS (Online Data Augmentation for Time Series Foundation Models), a principled strategy that generates synthetic data tailored to different training steps. OATS leverages valuable training samples as principled guiding signals and dynamically generates high-quality synthetic data conditioned on them. We further design a diffusion-based framework to produce realistic time series and introduce an explore-exploit mechanism to balance efficiency and effectiveness. Experiments on TSFMs demonstrate that OATS consistently outperforms regular training and yields substantial performance gains over static data augmentation baselines across six validation datasets and two TSFM architectures. The code is available at the link `https://anonymous.4open.science/r/OATS-536E`.

## 1 INTRODUCTION

Time series modeling plays a critical role across a wide range of domains, including finance (Kim et al., 2019; Li et al., 2024), healthcare (Guo et al., 2023), climate science (Liang et al., 2023), and industrial monitoring (Zamanzadeh Darban et al., 2024). Recent developments in time series foundation models (TSFMs) have further advanced this field by leveraging large-scale datasets collected from multiple third-party sources (Yao et al., 2024; Ansari et al., 2024), enabling cross-domain learning and zero-shot generalization. Nevertheless, the success of TSFMs relies on the availability of high-quality data. Typical challenges in time series datasets include missing values (Junninen et al., 2004), heterogeneous sampling rates (Woo et al., 2024), imbalanced domain distributions (Yao et al., 2024), data duplication (Lin et al., 2023). These issues make it more difficult to curate reliable large-scale time series datasets than in domains such as natural language processing (Gao et al., 2020; Raffel et al., 2020). As the community moves toward large-scale foundation models, synthetic data augmentation has emerged as a critical and widely adopted method for enhancing training data with synthetic samples. Leveraging the high controllability of time series patterns and operational simplicity, synthetic data augmentation can not only address data scarcity but also enriches domain diversity and improves the robustness of TSFMs (Liu et al., 2025).

Various data augmentation methods have been proposed in TSFM studies to enrich the training dataset with realistic and diverse synthetic data. The methods can be generally divided into two groups. One group of methods directly introduce *manually designed patterns* to generate new data samples. Such designed patterns include sinusoidal waves (Goswami et al., 2024), decomposed time series (Dooley et al., 2023) and handcrafted kernel bank (Shi et al., 2024; Ansari et al., 2024). Another group of methods utilize *basic transformations to existing time series data* and generate new ones. For example, smoothing, jittering (Um et al., 2017), and TSMixup (Ansari et al., 2024).

While these approaches have shown empirical effectiveness, they rely on *handcrafted heuristics* that are often *agnostic to the model training process*, which result in two critical challenges in designing high-performance data augmentation strategies. First, the quality of time series data for TSFMs is difficult to quantify in a principled way. The heuristics works for one time series tasks may fail for another (Liu et al., 2025; Kuvshinova et al., 2024). Second, recent studies show that the value of

the same data sample vary across the training process (Wang et al., 2024a), which challenges the capacity of the static data augmentation paradigm that generates synthetic data once and uniformly incorporated along the whole training process.

In this study, we address these challenges by leveraging recent advancements in *data attribution* (Koh & Liang, 2017; Deng et al., 2025) and *online data optimization* (Wang et al., 2024c). For the first challenge, data attribution aims to quantify the influence of individual training data points on model outputs; instead of studying heuristic rules like which pattern of time series data is helpful for the training, data attribution allows us to define and assess the quality of time series data by its influence on a utility function, e.g., validation loss, in a principle way. For the second challenge, we go beyond the static data augmentation paradigm in existing TSFM literature and conduct online data augmentation, which incorporates the training process information and dynamically generates high-quality data for each training step.

Concretely, we propose `OATS` (Online Data Augmentation for Time Series Foundation Models), a strategy to dynamically generate high-quality synthetic data in a principled manner. `OATS` generates high-quality synthetic data by using training samples with high data attribution scores (Koh & Liang, 2017) as guiding signals. `OATS` consists of three core components: **Time-series influence scores (TSIS)** integrate data attribution with time series–specific knowledge to dynamically assess the quality of each training sample in a principled manner to create generation guiding signal. **High-quality guided data augmentation** leverages the guiding signal to condition a diffusion model trained on a small subset of the TSFM training data for synthetic data generation. To reduce computational overhead and effectively balance between leveraging calculated scores and exploring new samples, `OATS` adopts an **explore–exploit mechanism**. Specifically, the influence scores are stochastically re-evaluated to incorporate model training dynamics (*"explore"*) while preserving previously identified high-quality data (*"exploit"*).

We summarize our contributions in this paper as follows:

- **An Online Data Augmentation Paradigm for TSFM**. We propose a new paradigm for TSFM training that generates synthetic data tailored to each training step. This approach expands the potential of the traditional static data augmentation paradigm by taking training process information into consideration.
- **A Novel Online Data Augmentation Strategy: `OATS`**. `OATS` leverages valuable training samples identified by data attribution scores as guiding signals and employs a diffusion model to generate synthetic data conditioned on these signals for each training step. Additionally, an explore-exploit mechanism is used to reduce computational cost and leverage local quality patterns. Overall, `OATS` offers a principled approach to dynamically generating high-quality synthetic data.
- **Comprehensive Empirical Evaluation**. We evaluate `OATS` on six evaluation datasets and two TSFM typical architectures (encoder-only and decoder-only TSFMs). `OATS` substantially outperform the baseline data augmentation methods as well as the regular training.

## 2 METHOD

In this section, we introduce the modules of Online Data Augmentation for Time Series Foundation Models (`OATS`). `OATS` consists of three modules, i.e., *Time-series Influence Scores (TSIS)* (Section 2.1) quantitatively estimate the quality of time series samples and to identify valuable data as guiding signal; *High-quality guided Data Generation* (Section 2.2) generates synthetic samples conditioned on the guiding signal; *Explore-exploit paradigm* (Section 2.3) balances the efficiency and effectiveness. We will introduce the modules in separate paragraphs. A diagram of the whole algorithm is presented in Figure 1 and an algorithm block is shown in Section 2.4.

**Set-up of online data augmentation in TSFM.** Suppose we have a training dataset $\mathcal{D}_{tr}$ of size $N$ which is partitioned into $L$ disjoint subsets: $\mathcal{D}_{tr} = \bigcup_{l=1}^{L} \mathcal{D}_l = \{z_{l,k} | l = [L]; k = [N_l]\}$, where $N_l = |D_l|$ is the size of subset $\mathcal{D}_l$ and $z_{l,k} \in \mathcal{Z}$, the data space of time series samples. We also have a validation dataset $\mathcal{D}_{val} = \{z_v | v = [N_v]\}$, $z_v \in \mathcal{Z}$ and $\mathcal{D}_{val}$ is not overlapped with $\mathcal{D}_{tr}$.

A TSFM parameterized by $w \in \mathcal{W}$ is being trained on $\mathcal{D}_{tr}$ to minimize loss function $\ell$ via an iterative optimization algorithm, e.g., stochastic gradient descent (Ruder, 2016) for $T$ steps. The intermediate checkpoints of each step are represented as $\{w_t | t = [T]\}$. At iteration $t$, a mini-batch

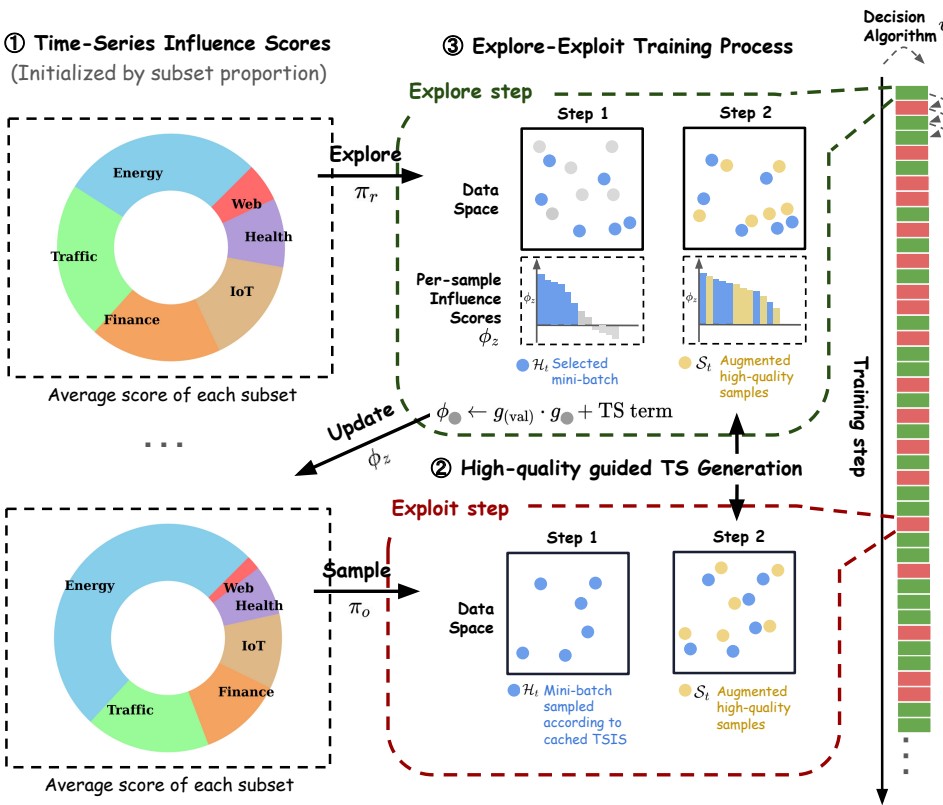

Figure 1: Architecture of `OATS`. `OATS` employs three modules: ① Time-Series Influence Scores (TSIS) create generation guiding signals as high-quality data samples and ② guide time series synthetic data generation for augmentation. ③ Explore-exploit mechanism comprehensively plans if updating the TSIS or leveraging cached scores.

of data $\mathcal{B}_t = \{z_1, z_2, \dots, z_B\}$ is sampled. Online data augmentation then generate synthetic samples $\mathcal{S}_t$, which are combine them with all or part of $\mathcal{B}_t$ to update the model.

## 2.1 CONSTRUCT GUIDING SIGNAL VIA TIME SERIES INFLUENCE SCORES

Generating synthetic data is widely used in the training process, yet a core question remains unsettled: *What should be generated?* A consistent and universally accepted answer has yet to emerge for the definition of the quality criteria for time series synthetic data in data augmentation. In `OATS`, we employ *data attribution* (Koh & Liang, 2017; Pruthi et al., 2020) as a principled method to estimate the influence of a data points to the model output, which has shown significant usefulness in dataset optimization in different areas. The data attribution scores to validation loss are taken as the quantitative quality indicator.

We design a time-series influence score (TSIS)[1] function with respect to a validation dataset $\mathcal{D}_{val}$ as $\mathcal{F}_{\mathcal{D}_{val}}$ for each sample to be $\mathcal{F}_{\mathcal{D}_{val}} : \mathcal{D}_{tr} \times \mathcal{W} \to \mathbb{R}$, which estimates the data attribution score of a data sample to be trained on $w_t$ with respect to the performance on $\mathcal{D}_{val}$. A larger score indicates that training on $z$ for the next training step $(t+1)$ leads to better performance on $\mathcal{D}_{val}$. The TSIS

---

[1]We will use "influence scores" and "data attribution scores" interchangeably.

function can be defined as:

$$\mathcal{F}_{\mathcal{D}_{val}}(z_i, w_t) = \underbrace{g_{(\text{val});\text{t}} \cdot g_{z_i;t}}_{\text{Influence Score}} - \underbrace{\mathbf{I}_{\text{SNR}(z_i)<k} \cdot \infty}_{\text{TS-specific quality}}, \tag{1}$$

where $g_{(\text{val});\text{t}} = \nabla_w \sum_{v_i \in \mathcal{D}_{val}} \ell(v_i, w_t)/|\mathcal{D}_{val}|$ and $g_{z_i;t} = \nabla_w \ell(z_i, w_t)$. The data attribution score is a first order Taylor approximation of the validation loss change, where a careful derivation is stated in Proposition 1. We additionally consider TS-specific quality indicators, which we use signal-to-noise ratio (SNR) in a pre-selection process, where $k$ is a threshold of minimal SNR and $\bar{\mathbf{I}}$ is the indicator function. High-quality data $\mathcal{H}_t$ is identified as the samples with top-$k$ TSIS scores in each mini-batch $\mathcal{B}_t$ of training step $t$, which will be used as guiding signals for synthetic data generation.

**Proposition 1** (First-order Taylor approximated influence score). *The influence score term of TSIS is a first-order Taylor approximation of the difference of a utility function before and after a training step. Typically, we define the utility function to be validation loss, which can be represented as*

$$\ell(\mathcal{D}_{val}, w_t) = \frac{1}{|\mathcal{D}_{val}|} \sum_{v_i \in \mathcal{D}_{val}} \ell(w_t, v_i)$$

$$\ell(\mathcal{D}_{val}, w_t) - \ell(\mathcal{D}_{val}, w_{t+1}) = (w_t - w_{t+1})^\top \nabla_w \ell(\mathcal{D}_{val}, w_t) + \mathcal{O}(\|w_{t+1} - w_t\|^2)$$

$$\simeq (w_t - w_{t+1})^\top \nabla_w \ell(\mathcal{D}_{val}, w_t)$$

$$= \eta_t \nabla_w \ell(z_i, w_t)^\top \nabla_w \ell(\mathcal{D}_{val}, w_t),$$

*where the model is updated through gradient descent[2] on a data sample $z_i$ with learning rate $\eta_t$, i.e.,*

$$w_{t+1} = w_t - \eta_t \nabla_w \ell(z_i, w_t).$$

*Given that the learning rate $\eta_t$ is typically small, the error of first-order Taylor approximation with level $\mathcal{O}(\|w_{t+1} - w_t\|^2)$ (or $\mathcal{O}(\eta^2)$ if the norm of gradient of loss is bounded) is small enough to provide an accurate estimate.*

## 2.2 HIGH-QUALITY DATA GUIDED DATA GENERATION

Once the guiding signal is constructed, a straight-forward method is *how can the synthetic data be generated?* We design a high-quality data guided generation model $\mathcal{G}$ for online synthetic data generation. We aim to model the conditional distribution $p(\hat{z}|\mathcal{H}_t)$, which defines the probability of generating a sample $\hat{z}$ given the guiding signal $\mathcal{H}_t \subseteq \mathcal{B}_t$ for each training step $t$. The conditional generation model $\mathcal{G}$ serves as a parameterized approximation of this distribution, enabling practical sampling via $\hat{z} \sim \mathcal{G}(\mathcal{H}_t)$. We design the architecture of $\mathcal{G}$ to be a diffusion model utilizing a time series semantic prototype module and take $\mathcal{H}_t$ as a generation condition. To incorporate the condition into the intermediate layers of noise prediction network, we take Huang et al. (2025) as our backbone model.

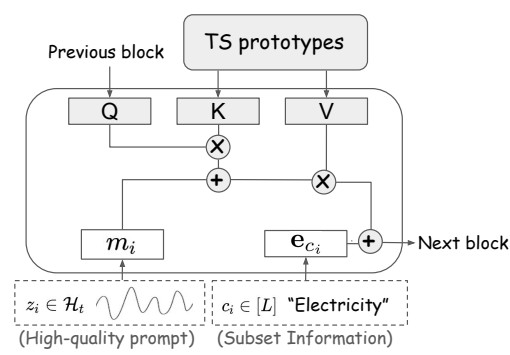

Figure 2: The architecture of the denoising diffusion model to generate synthetic data conditioned on constructed generation signals.

Our generation model $\mathcal{G}$ introduces a set of latent arrays $P$ as time series prototypes and extracts prototype weights $\mathcal{M} = \{m_i | z_i \in \mathcal{H}_t\}$, where we leverage the same design of weight extractor and time series prototypes initialization in Huang et al. (2025). We additionally include a class conditional guidance based on the subset information to enhance the condition, denoted as $\mathcal{C}_{\text{high}} = \{c_i | z_i \in \mathcal{H}_t\}$, where each $c_i \in [L]$ indicates which disjoint subset $\mathcal{D}_l$ the sample $z_i$ belongs to.

---

[2]Here we use SGD on a single data sample, which has been widely accepted as a reasonable approximation for other optimizers like Adam in data attribution studies(Wang et al., 2024c;b).

The generation (reverse diffusion) for each prompt can be expressed as

$$p(z^{(t-1)}|z^{(t)}, \mathcal{D}_{\text{high}}) = \mathcal{N}(z^{(t-1)}|\mu(z^{(t)}, t, \underbrace{m_i, c_i}_{\text{High}-\text{quality data guidance}}), \sigma_t), \quad (2)$$

where $z^{(t)}$ is the intermediate sample at diffusion timestep $t$, $\mathcal{N}$ is Gaussian distribution, $\sigma_t$ is the noise covariance at diffusion step $t$. The architecture of the parameterized conditional noise predictor incorporates the prototype weights $m_i$ and class condition $c_i$. We present the architecture of the high-quality guided data augmentation in Figure 2 in an intermediate $u^{\text{th}}$ U-Net layer. The formula can be represented as

$$h^{(u)} = \text{FF}(\text{Softmax}(\frac{QK^T}{\sqrt{d}} + m_i)V||\mathbf{e}_{c_i}), \quad (3)$$

where $Q = h^{(u-1)}W_Q^{(u)}$, $K = PW_K^{(u)}$ and $V = PW_V^{(u)}$ are the query, key and value for the attention. $W_Q \in \mathbb{R}^{d \times d}$, $W_K \in \mathbb{R}^{d \times d}$, $W_V \in \mathbb{R}^{d \times d}$ are learnable parameters, $P \in \mathbb{R}^{N_p \times d}$ is the prototype latent arrays, $\mathbf{e}_{c_i} \in \mathbb{R}^{d_c}$ is the embedding of class $c_i$, $d$ and $d_c$ is the hidden dimension of prototype embedding and class embedding, and $FF$ is a feed-forward network. The final U-Net layer's output $h^{\text{final}}$ is followed by another feed-forward network to produce the predicted noise.

## 2.3 EXPLORE & EXPLOIT MECHANISM

Ideally, $\mathcal{H}_t$ should be identified from the whole training dataset $\mathcal{D}_{tr}$ in Section 2.1, while the process of calculating TSIS for all training samples will be time-consuming. We are inspired by the multi-armed bandit problem and design an explore-exploit mechanism that could reuse the TSIS calculated for samples of previous training steps as well as explore new samples.

The mechanism divides training steps into *explore* steps and *exploit* steps. In *explore* steps, we calculate TSIS for a batch of data samples $\mathcal{B}_t$ sampled by strategy $\pi_r$. We assume the locality of TSIS among $z$ in the same subset. Based on this assumption, we maintain a dynamic cache $\Phi_{\mathcal{D}_l} \in \mathbb{R}, l \in [L]$ for each subset and update it through exponentially moving average. The partition of $\mathcal{D}_{tr} = \bigcup_{l=1}^{L} \mathcal{D}_l$ can naturally be the sub-datasets of a large collection or some clustering results. The update of $\Phi_{\mathcal{D}_l}$ on training step $t$ can be represented as

$$\Phi_{\mathcal{D}_l} = (1 - \beta)\Phi_{\mathcal{D}_l} + \beta \sum_{z_i \in \mathcal{D}_l \cap \mathcal{B}_t} \mathcal{F}_{\mathcal{D}_{val}}(z_i, w_t)/|\mathcal{D}_l \cap \mathcal{B}_t| \quad (4)$$

where $\mathcal{B}_t$ is a sample of size $|\mathcal{B}_t|$ from $\pi_r$, and $\pi_r$ is a uniform sampling strategy $\mathcal{U}(\mathcal{D}_{tr})$, $\beta \in [0, 1]$ is the hyperparameter controls the decay factor of the exponentially moving average. The design of $\pi_r$ for explore steps reflects the spirit to visit new data points and update the TS data. It randomly samples from the full training dataset $\mathcal{D}_{tr}$. Additionally, we refer to the influence score part of the TSIS in this subsection when we use $\mathcal{F}_{\mathcal{D}_{val}}$.

For *exploit* step, we design a sample algorithm $\pi_o$ utilizing the $\Phi_{\mathcal{D}_l}$ cached in *explore* step. The design of $\pi_o$ reflects the latest estimation of each subsets' quality.

$$\pi_o = \mathcal{U}(\mathcal{D}_l) \quad \text{with probability} \quad \frac{|\mathcal{D}_l| \cdot \max(0, \Phi_{\mathcal{D}_l})}{\sum_k |\mathcal{D}_k| \cdot \max(0, \Phi_{\mathcal{D}_k})} \quad (5)$$

The choice of *explore* or *exploit* step is handled by $\epsilon$-greedy, a popular strategy for explore and exploit. We design a strategy $\psi$ is defined to be "explore" with probability $\epsilon$ and "exploit" with probability $(1-\epsilon)$, where $\epsilon$ is a hyperparameter controlling the balance between explore and exploit. Notably, the overhead of *exploit* step is small enough to be ignored since TSIS is not calculated in such steps. A more careful complexity analysis is introduced in Appendix B.

## 2.4 ALGORITHM.

We summarize Online Data Augmentation for Time Series Foundation Models (OATS) and present in Algorithm 1.

---

**Algorithm 1** Online Data Augmentation for Time Series Foundation Models (`OATS`)

---

**Require:** Training dataset and its $L$ disjoint subsets $\mathcal{D}_{tr} = \bigcup_{l=1}^{L} \mathcal{D}_l$, validation dataset $\mathcal{D}_{val}$, training batch size $b$, conditional augmentation algorithm $\mathcal{G}$, training loss $\ell$, explore-exploit strategy $\psi$, explore sampling algorithm $\pi_r$, exploit sampling algorithm $\pi_o$, TSIS function $\mathcal{F}_{\mathcal{D}_{val}}$.

    Initialize model $w_0$
    Initialize TSIS per subset $\Phi_{\mathcal{D}_l}, l \in [L]$ to be subset proportion.
    **for** $t = 1$ to $T$ **do**
        **if** $\psi(t) ==$ "explore" **then**
            Sample $\mathcal{B}_t \sim \pi_r^b$                                        $\triangleright$ Step 1
            Calculate $\mathcal{F}_{\mathcal{D}_{val}}(z_i, w_t)$ for $z_i \in \mathcal{B}_t$
            Update $\Phi_{\mathcal{D}_l}$ according to Equation 4.
            Select a subset $\mathcal{H}_t \subseteq \mathcal{B}_t$ of samples with top-$b/2$ of value $\mathcal{F}_{\mathcal{D}_{val}}(z_i, w_t)$.
            Generate $b/2$ samples as $\mathcal{S}_t$ using $\mathcal{G}$ guided by $\mathcal{H}_t$.              $\triangleright$ Step 2
        **end if**
        **if** $\psi(t) ==$ "exploit" **then**
            Sample $\mathcal{H}_t \sim \pi_o^{b/2}$                                   $\triangleright$ Step 1
            Generate $b/2$ samples as $\mathcal{S}_t$ using $\mathcal{G}$ guided by $\mathcal{H}_t$.             $\triangleright$ Step 2
        **end if**
        Update $w_t$ on mini-batch data $\mathcal{H}_t \cup \mathcal{S}_t$ and get $w_{t+1}$, continue to next step $t + 1$.
    **end for**

---

## 3 EXPERIMENTS

In this section, we present the empirical evaluation of `OATS`. We first introduce the experiment setup in Section 3.1. We then evaluate the performance of `OATS` on typical TSFM architectures and datasets in Section 3.2. In addition, we examine the performance of `OATS` in different explore-exploit ratio in Section 3.3 and show some case studies in Section 3.4.

### 3.1 EXPERIMENT SETUP

**Model.** We conduct experiments on two typical TSFM architectures, i.e., encoder-only and decoder-only Transformer. We follow the same training settings and model definitions, which incorporate patch embedding, rotary positional embedding, and a mixture of distributions to better adapt to time series forecasting while preserving extensibility, in Yao et al. (2024) to match the popular TSFM forecasting models, Moirai (Woo et al., 2024) and Chronos (Ansari et al., 2024). We will call these two models "Encoder-only TSFM" and "Decoder-only TSFM" in our following experiments.

**Datasets.** Following Yao et al. (2024), we pre-train "Encoder-only TSFM" and "Decoder-only TSFM" on the LOTSA dataset in pretraining stage. These models are then evaluated on the out-of-distribution LSF dataset (Wu et al., 2023), using various prediction length and prepossessing pipeline as in Yao et al. (2024). We evaluate 6 out-of-distribution datasets in LSF (Wu et al., 2023) (ETTm1, ETTm2, ETTh1, ETTh2, Weather, Electricity). We also take a very small number of samples (32 in all our experiments) from these evaluation datasets as validation set used by TSIS. The training dataset of the high-quality prompt guided diffusion model is a very small subset sampled from the training dataset of TSFM. In our experiment, we sample 5% of the training data.

**Baselines.** We examine two popular data augmentation methods for TSFM. "TSMixup" or Time Series Mixup is proposed in Ansari et al. (2024) which creates new data samples from $k$ existing ones through weighted summation. The generation process can be represented as $\hat{z} = \sum_{i=1}^{k} \lambda_i z_i$, where $\hat{z}$ is the generated sample and $\lambda_i$ is the weight for the $i^{\text{th}}$ sample. "Jitter" (Um et al., 2017) is another widely adopted data augmentation method that involve small noise on existing samples to increase the robustness and diversity. The generation process can be represented as $\hat{z} = z + \epsilon$, where $\epsilon \sim \mathcal{N}(0, \sigma)$ and $\sigma$ is the noise variance. We also include the result of the regular training process without additional data augmentation.

**Evaluation Metrics.** To be consistent with Yao et al. (2024), we primarily report the normalized mean absolute percentage error (MAPE) and negative log-likelihood (NLL) for Encoder-only TSFM and Decoder-only TSFM, as these metrics avoid distortions caused by high-amplitude samples. We

specifically report the metrics for prediction length to be 192 and "overall" (average performance of using prediction lengths).

## 3.2 PERFORMANCE OF OATS

In Table 1 and Table 2, we present the NLL and MAPE on the test datasets on Encoder-only TSFM and Decoder-only TSFM respectively. OATS outperforms baselines as well as regular training in almost all cases and both metrics. Especially, for datasets ETTm1, ETTm2, ETTh2, Weather and Electricity, OATS reach the best result. TSMixup and Jitter get mixed performance compared to regular training, as presented by the light green and red background. While OATS performs more consistently better than regular training. In Figure 3, we present the test loss (NLL) curve on evaluation datasets on Encoder-only TSFM and different test datasets. OATS achieves a faster reduction in NLL than other baseline data augmentation methods as well as the regular training and achieves better overall performance.

Table 1: Performance comparison of various data augmentation methods on encoder-only TSFM with prediction length be 192 / average over all prediction lengths) and $\epsilon = 1$. **Bold** means the best result. Light green background means that the performance is better than regular training process, while light red background means that the performance is worse than regular training process. The error bar shows the standard error of mean over 5 independent runs.

| Dataset | Pred. length | OATS | | TSMixup | | Jitter | | Regular | |
|---|---|---|---|---|---|---|---|---|---|
| | | NLL | MAPE | NLL | MAPE | NLL | MAPE | NLL | MAPE |
| ETTm1 | 192 | **1.627 ± 0.042** | **0.672 ± 0.043** | 1.725 ± 0.031 | 0.759 ± 0.038 | 1.715 ± 0.047 | 0.691 ± 0.044 | 1.870 ± 0.019 | 0.844 ± 0.056 |
| | Overall | **1.614 ± 0.020** | **0.623 ± 0.024** | 1.715 ± 0.018 | 0.700 ± 0.024 | 1.731 ± 0.025 | 0.650 ± 0.019 | 1.854 ± 0.016 | 0.783 ± 0.037 |
| ETTm2 | 192 | **1.872 ± 0.014** | **0.208 ± 0.005** | 2.083 ± 0.016 | 0.271 ± 0.006 | 2.068 ± 0.020 | 0.256 ± 0.007 | 2.118 ± 0.028 | 0.267 ± 0.009 |
| | Overall | **1.863 ± 0.013** | **0.222 ± 0.004** | 2.097 ± 0.011 | 0.292 ± 0.003 | 2.072 ± 0.009 | 0.273 ± 0.005 | 2.128 ± 0.014 | 0.290 ± 0.004 |
| ETTh1 | 192 | **1.794 ± 0.035** | 0.681 ± 0.025 | 1.883 ± 0.030 | 0.634 ± 0.025 | 1.959 ± 0.028 | **0.629 ± 0.035** | 1.897 ± 0.040 | 0.758 ± 0.059 |
| | Overall | **1.814 ± 0.020** | 0.709 ± 0.021 | 1.860 ± 0.027 | **0.634 ± 0.011** | 1.913 ± 0.026 | 0.644 ± 0.021 | 1.915 ± 0.019 | 0.812 ± 0.031 |
| ETTh2 | 192 | **1.857 ± 0.063** | **0.267 ± 0.007** | 2.123 ± 0.029 | 0.294 ± 0.011 | 2.127 ± 0.022 | 0.296 ± 0.013 | 2.084 ± 0.047 | 0.282 ± 0.015 |
| | Overall | **1.876 ± 0.025** | **0.263 ± 0.005** | 2.113 ± 0.020 | 0.294 ± 0.005 | 2.122 ± 0.022 | 0.298 ± 0.005 | 2.086 ± 0.021 | 0.283 ± 0.008 |
| Weather | 192 | **3.193 ± 0.043** | **1.778 ± 0.127** | 3.413 ± 0.041 | 2.016 ± 0.040 | 3.460 ± 0.043 | 2.619 ± 0.280 | 3.486 ± 0.051 | 2.331 ± 0.285 |
| | Overall | **3.216 ± 0.036** | **1.659 ± 0.069** | 3.428 ± 0.023 | 1.948 ± 0.033 | 3.499 ± 0.020 | 2.591 ± 0.145 | 3.526 ± 0.026 | 2.267 ± 0.125 |
| Electricity | 192 | **5.967 ± 0.021** | **0.515 ± 0.071** | 5.983 ± 0.010 | 0.553 ± 0.034 | 6.050 ± 0.015 | 0.659 ± 0.062 | 6.266 ± 0.019 | 0.750 ± 0.033 |
| | Overall | **5.945 ± 0.011** | **0.507 ± 0.045** | 5.987 ± 0.007 | 0.555 ± 0.013 | 6.044 ± 0.010 | 0.642 ± 0.025 | 6.260 ± 0.015 | 0.767 ± 0.021 |

Table 2: Performance comparison of various data augmentation methods on decoder-only TSFM with prediction length be 192 / average over all prediction lengths) and $\epsilon = 1$. **Bold** means the best result. Light green background means that the performance is better than regular training process, while light red background means that the performance is worse than regular training process. The error bar shows the standard error of mean over 5 independent runs..

| Dataset | Pred. length | OATS | | TSMixup | | Jitter | | Regular | |
|---|---|---|---|---|---|---|---|---|---|
| | | NLL | MAPE | NLL | MAPE | NLL | MAPE | NLL | MAPE |
| ETTm1 | 192 | **1.654 ± 0.018** | **0.659 ± 0.027** | 1.691 ± 0.013 | 0.664 ± 0.027 | 1.767 ± 0.041 | 0.675 ± 0.031 | 1.737 ± 0.034 | 0.677 ± 0.022 |
| | Overall | **1.740 ± 0.015** | **0.635 ± 0.014** | 1.785 ± 0.025 | 0.649 ± 0.016 | 1.842 ± 0.034 | 0.650 ± 0.018 | 1.812 ± 0.022 | 0.653 ± 0.015 |
| ETTm2 | 192 | **1.726 ± 0.039** | **0.202 ± 0.003** | 1.789 ± 0.014 | 0.206 ± 0.002 | 1.857 ± 0.044 | 0.222 ± 0.007 | 1.841 ± 0.009 | 0.217 ± 0.001 |
| | Overall | **1.765 ± 0.030** | **0.228 ± 0.002** | 1.824 ± 0.015 | 0.229 ± 0.002 | 1.880 ± 0.034 | 0.244 ± 0.008 | 1.874 ± 0.008 | 0.242 ± 0.004 |
| ETTh1 | 192 | **1.759 ± 0.046** | 0.534 ± 0.022 | 1.933 ± 0.069 | 0.536 ± 0.017 | 1.808 ± 0.069 | **0.504 ± 0.021** | 1.852 ± 0.046 | 0.537 ± 0.026 |
| | Overall | **1.824 ± 0.020** | 0.562 ± 0.022 | 2.038 ± 0.031 | 0.568 ± 0.014 | 1.878 ± 0.030 | **0.538 ± 0.016** | 1.943 ± 0.025 | 0.563 ± 0.013 |
| ETTh2 | 192 | **1.817 ± 0.045** | **0.261 ± 0.011** | 1.977 ± 0.036 | 0.289 ± 0.009 | 1.923 ± 0.036 | 0.275 ± 0.014 | 1.890 ± 0.050 | 0.284 ± 0.008 |
| | Overall | **1.936 ± 0.024** | **0.274 ± 0.007** | 2.087 ± 0.030 | 0.307 ± 0.005 | 1.994 ± 0.022 | 0.295 ± 0.008 | 2.027 ± 0.023 | 0.309 ± 0.006 |
| Weather | 192 | **2.914 ± 0.081** | **2.885 ± 0.650** | 2.974 ± 0.059 | 3.935 ± 0.334 | 3.025 ± 0.065 | 5.423 ± 0.822 | 2.914 ± 0.026 | 3.059 ± 0.622 |
| | Overall | **3.168 ± 0.058** | **2.376 ± 0.323** | 3.265 ± 0.039 | 3.643 ± 0.186 | 3.276 ± 0.037 | 4.624 ± 0.411 | 3.245 ± 0.041 | 2.946 ± 0.574 |
| Electricity | 192 | **6.041 ± 0.017** | **0.528 ± 0.030** | 6.074 ± 0.033 | 0.563 ± 0.010 | 6.090 ± 0.022 | 0.627 ± 0.028 | 6.049 ± 0.026 | 0.588 ± 0.020 |
| | Overall | **6.040 ± 0.011** | **0.526 ± 0.014** | 6.079 ± 0.018 | 0.564 ± 0.014 | 6.107 ± 0.019 | 0.631 ± 0.018 | 6.057 ± 0.015 | 0.579 ± 0.011 |

## 3.3 PERFORMANCE OF OATS UNDER DIFFERENT EXPLORE-EXPLOIT LEVELS

The key experiment goal of this subsection is examine the sensitivity and behavior of OATS on different levels of explore-exploit mechanisms through adjusting the value of $\epsilon$. $\epsilon$ is a hyperparameter designed to control the possibility of carrying out *explore* ($\epsilon$) or *exploit* step ($1 - \epsilon$). *Explore* step

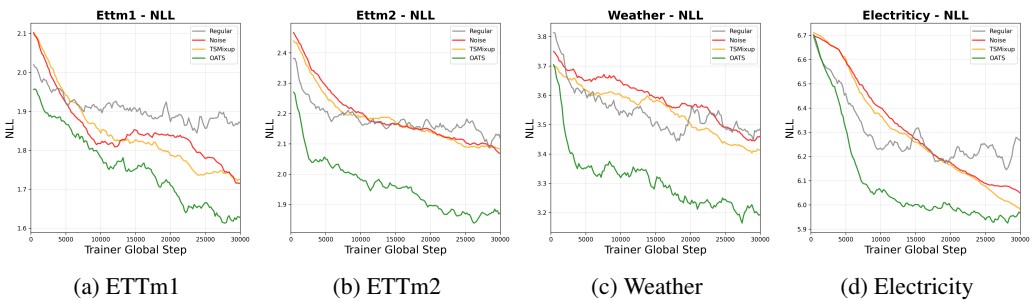

Figure 3: Test loss (NLL) of `OATS`, TSMixup, Jitter and Regular training for each training step.

will examine new data by calculating the TSIS ($\mathcal{F}_{\mathcal{D}_{val}}$), which is time-consuming. *Exploit* step will leverage cached TSIS ($\Phi_{\mathcal{D}_l}$) calculated in previous steps to directly sample high-quality batch, which is light-weighted.

In Figure 4 we present the performance of `OATS` on different levels of explore-exploit mechanisms. We choose to set $\epsilon = [0.3, 0.5, 0.7, 1.0]$ and report the performance on two test datasets (ETTh1 and Electricity) and four prediction lengths (96, 192, 336, 720).

The best performance in each setting is often not achieved by $\epsilon = 1$ (all explore step), the most computational heavy setting that calculate TSIS in every step. This shows that leveraging cached TSIS is helpful to select more high-quality batch (using $\pi_o$) to guide the data augmentation and demonstrates the potential of using the mechanism to reduce computational overhead. Another observation also shows that `OATS` consistently performs better than regular training process regardless the setting of $\epsilon$. We included a detailed complexity and time cost analysis in Appendix B.

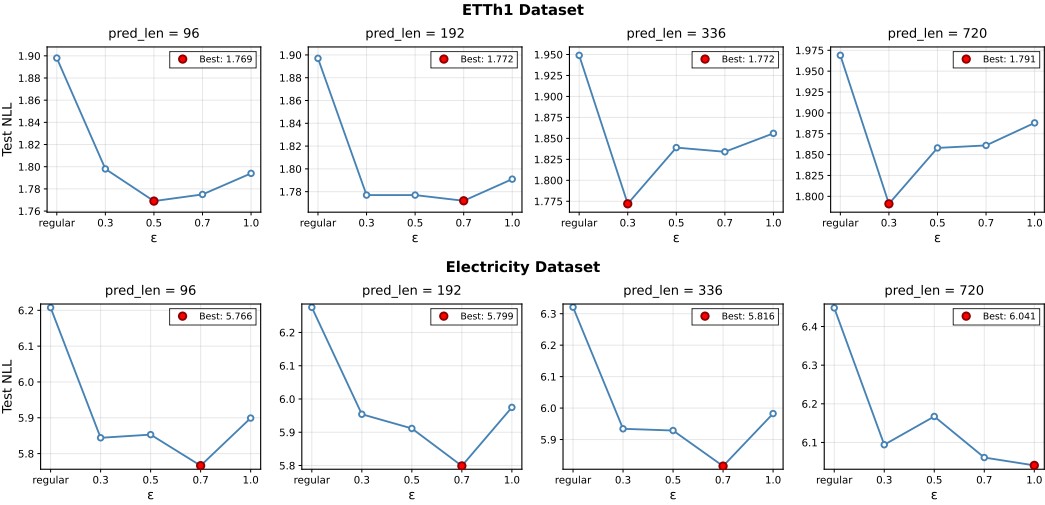

Figure 4: Performance of `OATS` on different explore-exploit ratio $\epsilon$. (**First Row**) Test NLL on ETTh1. (**Second Row**) Test NLL on Electricity. The best performance is labeled by red circle.

### 3.4 CASE STUDIES

In this subsection, we carry out two case studies to further provide intuitive evidences of our motivation and some insights.

**Contribution of each sub-dataset.** In the first case study, we present the per–sub-dataset TSIS, i.e., $\Phi_{\mathcal{D}_l}$, throughout the training process (Figure 5a), along with the original proportion of each sub-dataset in the training corpus (Figure 5b). The results reveal two key insights: (1) the contribution of each sub-dataset evolves during training, which intuitively motivates the use of online data

augmentation; and (2) the contribution of a training sub-dataset does not necessarily align with its size. For instance, a relatively small sub-dataset (e.g., solar_power) can contribute substantially to reducing test loss, whereas some large sub-datasets (e.g., those grouped under others, which account for 60% of the total data in Figure 5b) provide only limited contribution (around 10% in Figure 5a). Such findings are difficult to uncover through heuristic design or empirical intuition alone, but could be revealed through principled influence analysis.

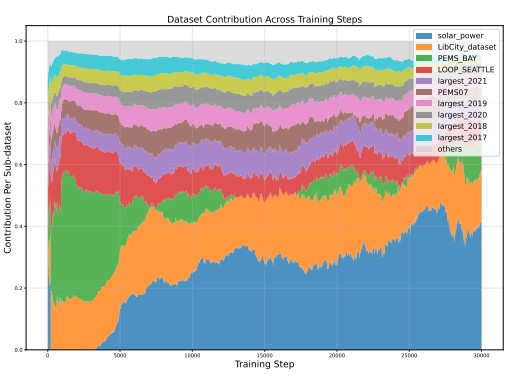
(a) Sub-dataset TSIS ($\Phi_{\mathcal{D}_l}$) along the training.

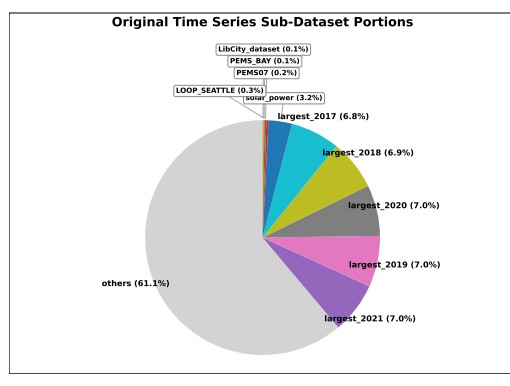
(b) Sub-dataset portion in training dataset.

Figure 5: Comparison between sub-dataset data contribution and data portion.

**Selected high-quality prompts and generated samples.** In the second case study, we present the high-quality data samples prompts selected according to TSIS and the corresponded guided generation in Figure 6. The results shows that the generation model of the high-quality guided data augmentation could generate realistic data samples similar to the prompt with various patterns.

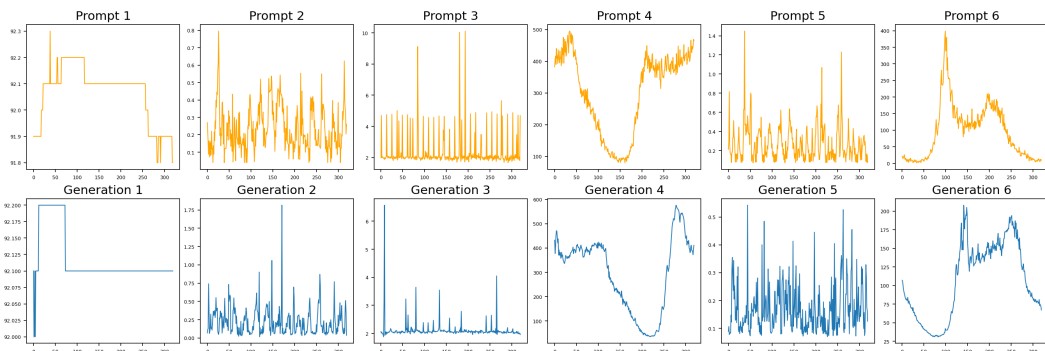

Figure 6: (**First row**) High-quality data samples prompt. (**Second row**) The corresponded guided generation.

## 4 CONCLUSION

In this paper, we introduce OATS, a principled strategy for generating synthetic data tailored to different training steps in TSFMs. The core idea behind OATS is to leverages valuable training samples as principled guiding signals and dynamically generates high-quality synthetic data conditioned on them. To further improve efficacy and efficiency, we propose an explore–exploit mechanism that leverages cached influence scores to reduce computational overhead. Empirical evaluations across diverse model architectures, datasets, and prediction lengths demonstrate that OATS substantially outperforms both standard data augmentation methods and regular training in terms of TSFM performance. By expanding the scope of online TSFM data augmentation, OATS enables researchers to systematically optimize TSFM training datasets in a principle way.

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

## A   RELATED WORKS

**Data Augmentation in TSFMs.**   Various data augmentation methods have been proposed in TSFM studies to enrich the training dataset with realistic and diverse synthetic data. The methods can be generally divided into two groups. One group of methods directly introduces manually designed patterns to generate new data samples. For example, Moment (Goswami et al., 2024) directly introduces sinusoidal waves with varying frequencies as augmented training data. ForecastPFN (Dooley et al., 2023) employs time series decomposition and generates each component (e.g., trend, seasonality, and noise) using hand-selected hyperparameters. TimesFM (Das et al., 2024) incorporates additional statistical patterns such as ARMA processes and piecewise linear trends to ensure the synthetic data is not overly simplistic. Some works (Shi et al., 2024; Ansari et al., 2024) adopt KernelSynth, which generates synthetic data from a handcrafted kernel bank designed to produce "high-quality" samples under human-defined constraints. Another group of methods utilizes basic transformations to existing time series data and generates new ones. For example, smoothing and jittering (Um et al., 2017) are used to involve diversity through various levels of noise. Ansari et al. (2024) proposes TSMixup to generate new samples through weighted summation of several existing ones. While these approaches enrich the training data and have shown practical effectiveness, they often rely on carefully crafted heuristic choices made before training, leaving an open question of how to identify a high-performance data augmentation strategy in a more general and principled way. Notably, Taga et al. propose an online data augmentation algorithm that combines a RHO-based selection method with traditional time series transformations such as jittering, which shares similarities with our approach. However, their simplistic synthetic data generation method may lead to performance degradation. Furthermore, the RHO-based method incurs substantial computational costs due to its requirement for training a reference model.

**Training Data Attribution.**   Training data attribution aims to quantitatively estimate the influence of each training example on model behavior. This research direction traces back to robust statistics in the 1980s and has recently gained renewed attention with the use of influence functions to interpret black-box neural networks (Koh & Liang, 2017). More recent work has focused on improving both the accuracy (Wang et al., 2024b; Ilyas et al., 2022) and scalability (Park et al., 2023; Choe et al., 2024; Grosse et al., 2023) of attribution methods. These approaches have also been applied in practical scenarios such as data selection (Wang et al., 2024c) and data valuation for data markets (Deng & Ma, 2023).

**Time Series Generation.**   Time series generation has been explored using a variety of model architectures. GAN-based approaches train adversarial objectives to generate realistic sequences (Yoon et al., 2019). VAE-based methods design specialized decoders tailored for time series (Desai et al., 2021). More recently, diffusion-based methods have been developed, leveraging different denoising architectures. Beyond unconditional synthesis, conditional generation has been studied in settings such as metadata-conditioned generation (Narasimhan et al., 2024) and prompt-based generation (Huang et al., 2025).

## B   COMPLEXITY ANALYSIS

`OATS` involves computational overhead to the training process, which can be spitted into two sources: 1) calculation of TSIS and 2) synthetic data generation. **In summary**, the overhead of TSIS calculation is up to $\epsilon \times$ of the regular training step. The good performance shown in Figure 4 with $\epsilon = 0.3$ shows that this overhead could be relatively small. The overhead of synthetic data generation model is a fixed number, which gets lower as the regular training step has higher time cost, e.g., a larger TSFM with more parameters.

**Calculation of TSIS**   For TSIS calculation, the overhead of Equation 1 mainly comes from the gradient calculation of validation data samples in "explore" steps. In our experiment, we use a small number of validation data samples (32) that matches the training batch size. This helps maintain the same computation complexity between TSIS and the regular gradient descent training step as $\mathcal{O}(b)$, where $b$ is the training batch size. Additionally, we also adopted an efficient implementation called "Ghost Inner Product" (Wang et al., 2024a) to avoid per-sample gradient calculation while get the dot product result.

**Lemma 1** (Ghost Inner Product). *In the influence function calculation in TSIS, which includes sample-level gradient dot products, i.e., $\nabla_w \ell(z^{(1)}, w_t) \cdot \nabla_w \ell(z^{(2)}, w_t)$. We may only perform the backpropagation once and calculate the gradient dot product. Ghost Inner Product perform the product layer by layer in a model and here we present the method for a simple linear layer.*

$$\frac{\partial \ell^{(1)}}{\partial w} \cdot \frac{\partial \ell^{(2)}}{\partial w} = (a^{(1)} \otimes \frac{\partial \ell^{(1)}}{\partial s^{(1)}}) \cdot (a^{(2)} \otimes \frac{\partial \ell^{(2)}}{\partial s^{(2)}}) = ((a^{(1)})^\top a^{(2)})((\frac{\partial \ell^{(1)}}{\partial s^{(1)}})^\top \frac{\partial \ell^{(2)}}{\partial s^{(2)}}), \quad (6)$$

*where $a^{(1)}$ and $a^{(2)}$ is the linear layer's input, $s^{(1)}$ and $s^{(2)}$ is the pre-activation output, and $\otimes$ indicates the outer product.*

In TSIS, $\ell^{(1)} = \ell(z_i, w_t)$ in each training step $t$ and $z_i \in \mathcal{B}_t$ and $\ell^{(2)} = \ell(\mathcal{D}_{val}, w_t)$. "Ghost Inner Product" enable the implementation of influence score in TSIS to carry out **only one additional backpropagation** on $\ell(\mathcal{B}_t, w_t) + \ell(\mathcal{D}_{val}, w_t)$ to get all terms in Equation 6 compared to regular training.

Furthermore, the overhead of "exploit" steps is small enough to be ignored since TSIS in Equation 1 is not used in those steps, **which means that the computational overhead of TSIS will be reduced by (1-$\epsilon$) and memory footprint will be on par with the regular training**. In Figure 4, $\epsilon = 0.3$ also substantially outperform regular training process, which shows good potential of reducing overhead. Following table 3 is a time cost record for training batch size 32. It is worth noting that the actual implementation may affect the time cost.

Table 3: Per-training step time cost under different settings of Explore-exploit ratio $\epsilon$.

| $\epsilon$ | Regular | 0.3 | 0.5 | 0.7 | 1.0 |
|---|---|---|---|---|---|
| Avg. time cost per step (s) on A40 GPU | 0.74 | 0.95 | 1.09 | 1.20 | 1.32 |

**Synthetic data generation.** For synthetic data generation through diffusion model sampling, it is hard to compare the complexity with the regular gradient descent training step. Empirically, we find that some accelerated sampling strategies like DDIM (Song et al., 2020) could perform well enough with very small sample steps. Moreover, the geneartion overhead will be a fixed term and the relative proportion will get lower as the TSFMs get larger, e.g., with more parameters. The overhead runtime of current synthetic data generation is 0.022s per synthetic sample, which is $0.022 \times 32 \times 0.5 = 0.35s$ additional to the Table 3.

**Empirical results at same training time.** We also show the performance of OATS and baselines' performance under the **same training time as regular training**. OATS still outperform other baselines. It is also worth noting that with a limited training time, OATS may be more far away from converging.

Table 4: Performance (NLL) of OATS and baselines' performance under the same time cost.

| Dataset | OATS($\epsilon$=0.3) | OATS($\epsilon$=0.5) | OATS($\epsilon$=0.7) | OATS($\epsilon$=1) | TSMixup | Jitter | Regular |
|---|---|---|---|---|---|---|---|
| Electricity | **6.00** | 6.11 | 6.02 | 6.04 | 6.15 | 6.14 | 6.26 |
| ETTh1 | 1.82 | **1.81** | 1.85 | 1.83 | 1.88 | 1.94 | 1.89 |

## C EXPERIMENT SETTINGS

**TSFM Models.** We conduct the experiment on the TSFM models with the same configuration as the encoder-only model with a 10M parameter size in Yao et al. (2024). The model is a modified version of Moirai (Woo et al., 2024), introduced by Yao et al. (2024), which incorporates patch embedding, rotary positional embedding, and a mixture of distributions to better adapt to time series forecasting while preserving extensibility.

**TSFM Training Process.** We adopt a similar training setting as in Yao et al. (2024) for the experiment. We utilize the AdamW optimizer with a batch size of 32 and a maximum learning rate of $10^{-3}$ with a linear warm-up of $10^4$ training steps, followed by cosine decay for the remaining $2 \times 10^4$ steps.

**Datasets for TSFM.** We pretrain the TSFM on the modified (balanced domain sample, quality filtering) LOTSA-100M dataset in the pretraining stage provided by Yao et al. (2024). We take the native sub-dataset division as subsets in our experiment. These models are then evaluated on the out-of-distribution LSF dataset (Wu et al., 2023), using various prediction lengths (96, 192, 336) and the same preprocessing pipeline as in Yao et al. (2024). The detailed information of evaluation datasets is stated in Table 5.

Table 5: Evaluation datasets and properties.

| Dataset | Domain | Frequency | # Prediction Length |
|---|---|---|---|
| ETTh1 | Energy | H | 96/192/336/720 |
| ETTh2 | Energy | H | 96/192/336/720 |
| ETTm1 | Energy | 15min | 96/192/336/720 |
| ETTm2 | Energy | 15min | 96/192/336/720 |
| Electricity | Energy | H | 96/192/336/720 |
| Weather | Climate | H | 96/192/336/720 |

**Generation Model.** We leverage the architecture in Huang et al. (2025) as the backbone model. The denoising network of the diffusion model employed adopts a U-Net architecture comprising 4 up/down sampling blocks with 8 attention heads and the dimension of each head is 64. An additional class embedding with 64 dimension is added to the output of each block.

**Generation Model Training Process.** We train a Latent Diffusion Model for time series generation using 200 diffusion time steps with a linear noise schedule ranging from 0.0005 to 0.1. The model employs L1 loss as the training objective and a dropout probability of 0.5 to enable classifier-free guidance. We set the base learning rate to 0.001.

**Datasets for Generation Model.** We train the generation model for high-quality guided data augmentation described in Section 2 by a sampled dataset from the training dataset of TSFM. We sample 5% of the dataset in 20 selected subsets (Table 6) in LOTSA-100M as the training set of the diffusion model. The generation length of the diffusion model is set to 320, and the diffusion step is set to 200. We use DDIM for the sampling process with 20 steps.

**Hyperparameters.** For experiments in Table 1 and Table 2, we set hyperparameters of TSMixup baseline to have $k = 2$ and $\lambda_0 \sim \mathcal{U}(0.1, 0.9)$, $\lambda_1 = 1 - \lambda_0$. For Jitter, we use $\sigma = 0.03$. For `OATS`, we fixed $\epsilon = 1$ in Table 1 and Table 2. Experiments with $\epsilon < 1$ will be included in Figure 4.

## D ADDITIONAL RESULTS

In Table 7, we present an ablation study between `OATS` and a partially implemented "`OATS` (Sel only)" that only trains on the selected guiding signals. Essentially, "`OATS` (Sel only)" is Algorithm 1 without "Step 2" in both explore and exploit steps. The results shows that `OATS` outperform the partially implementation in all settings. The experiment is carried out on Encoder-only TSFM and "ETT" refers to ETTm2.

In Figure 7, we present more examples between high-quality data sample prompts and the corresponding generated samples.

Table 6: Training datasets of generation model.

| Dataset | Domain | Frequency |
|---|---|---|
| CMIP6 | Climate | 6H |
| ERA5 | Climate | H |
| CloudOpsTSF | CloudOTS | 5T |
| Azure VM Traces 2017 | CloudOTS | 5T |
| Loop Seattle | Transport | 5T |
| PEMS07 | Transport | 5T |
| PEMS Bay | Transport | 5T |
| Q-Traffic | Transport | 15T |
| Largest 2017 | Transport | 5T |
| Largest 2018 | Transport | 5T |
| Largest 2019 | Transport | 5T |
| Largest 2020 | Transport | 5T |
| Largest 2021 | Transport | 5T |
| Australian Electricity | Energy | 30T |
| Buildings900K | Energy | H |
| Solar Power | Energy | 4S |
| Favorita Sales | Sales | D |
| Wiki-Rolling | Web | D |
| LibCity | Transport | 5T |
| OthersLOTSA | Energy | H |

Table 7: Ablation study of the high-quality data selection step.

| Dataset | Pred. length | OATS | | OATS (Sel only) | | Regular | |
|---|---|---|---|---|---|---|---|
| | | NLL | MAPE | NLL | MAPE | NLL | MAPE |
| ETT | 96 | **1.879** | **0.216** | 2.156 | 0.303 | 2.177 | 0.306 |
| | 192 | **1.872** | **0.208** | 2.106 | 0.274 | 2.124 | 0.271 |
| | 336 | **1.839** | **0.242** | 2.051 | 0.305 | 2.095 | 0.298 |
| | 720 | **1.938** | **0.284** | 2.123 | 0.337 | 2.204 | 0.328 |
| Electricity | 96 | **5.882** | **0.425** | 6.091 | 0.649 | 6.187 | 0.731 |
| | 192 | **5.967** | **0.515** | 6.147 | 0.632 | 6.244 | 0.744 |
| | 336 | **5.985** | **0.582** | 6.229 | 0.664 | 6.284 | 0.810 |
| | 720 | **6.053** | **0.580** | 6.345 | 0.674 | 6.411 | 0.822 |
| Weather | 96 | **3.192** | **1.498** | 3.456 | 2.305 | 3.522 | 2.347 |
| | 192 | **3.193** | **1.778** | 3.464 | 2.661 | 3.485 | 2.303 |
| | 336 | **3.263** | **1.702** | 3.536 | 2.630 | 3.582 | 2.125 |
| | 720 | **3.675** | **1.889** | 3.916 | 2.638 | 4.016 | 1.991 |

### D.1 IMPROVEMENT OF CLASS EMBEDDING ON CONDITIONAL GENERATION.

We also include Figure 8 an Figure 9 show the improvement of class embedding. In Figure 8, the generation (the second to last of each row) of model without the class embedding show broken and unaligned pattern with the prompt (the first one for each row). Figure 9, on the other hand, shows that class embedding helps the generation to align with the prompt in pattern.

### D.2 ADDITIONAL DATA-DRIVEN BASELINE

We add a baseline using data-driven generative models (diffusion model, termed as "DD") to generate synthetic data in an offline paradigm. The results in Table 8 show that OATS could significantly perform better than the all baselines. The unconditional diffusion model is trained on the same data shown in Table 6 with 4 sampling blocks with 8 attention heads and the same training process stated in Appendix C.

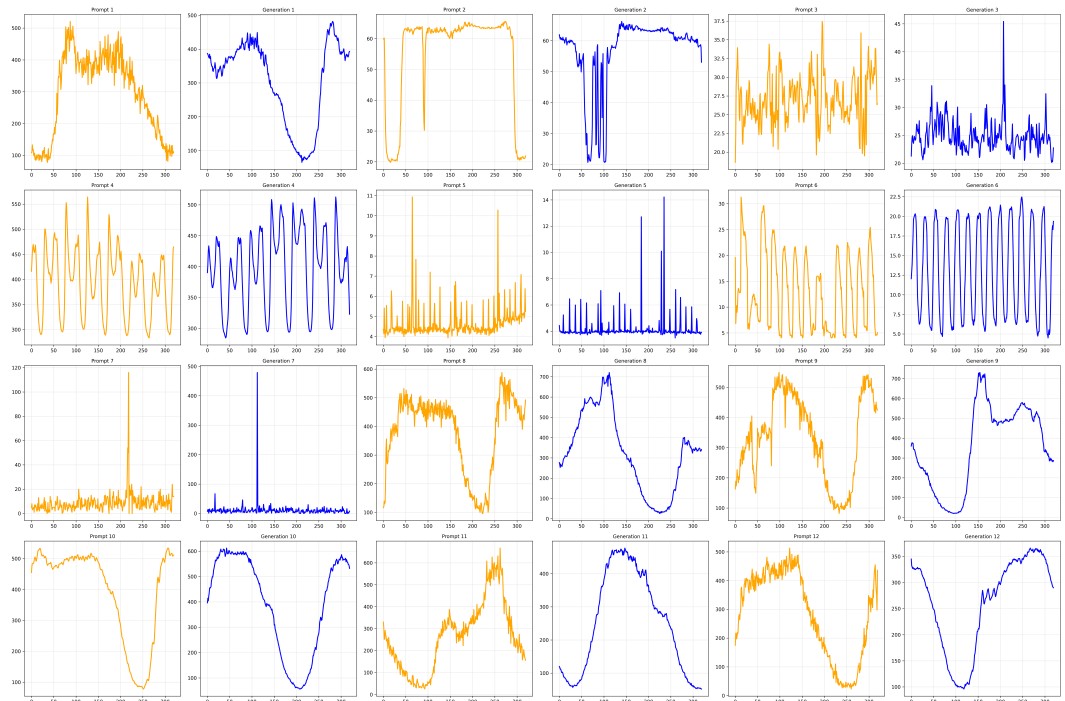

Figure 7: More examples of high-quality data samples prompts (yellow) and the corresponded guided generation (blue).

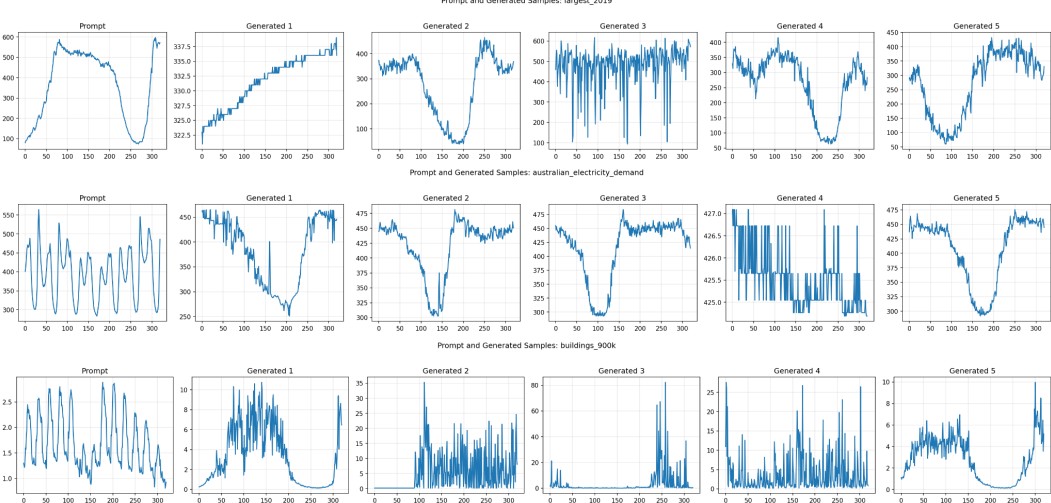

Figure 8: Examples of data generation of diffusion model **without** class embedding. For each row, the first sample is prompt and the following five are guided generations.

## D.3 SENSITIVITY OF INITIAL PARTITION

In Table 9, we follow the native partition structure (sub-datasets) provided in the LOTSA dataset and evaluate different settings of initial data granularity. "Native Subdataset" refers to the default setting used in the paper, where we maintain the original partition structure. "Combine each two" represents a coarser granularity setting where we merge two native subdatasets into a single partition. Additionally, we include a baseline with random partitioning, where data samples are randomly

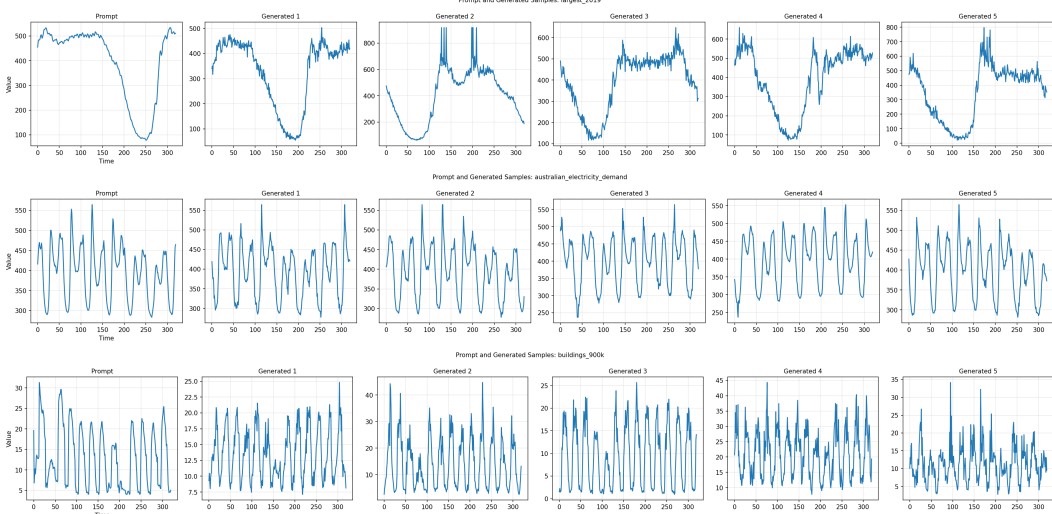

Figure 9: Examples of data generation of diffusion model **with** class embedding. For each row, the first sample is prompt and the following five are guided generations.

Table 8: Performance comparison of different methods with error bars. **Bold** means the best result. Light green background means that the performance is better than regular training process, while light red background means that the performance is worse than regular training process. The error bar shows the standard error of mean over 5 independent runs.

| Dataset | Pred. length | OATS | | DD | | Jitter | | TSMixUp | | Regular | |
|---|---|---|---|---|---|---|---|---|---|---|---|
| | | NLL | MAPE | NLL | MAPE | NLL | MAPE | NLL | MAPE | NLL | MAPE |
| ETTm1 | 96 | **1.557 ± 0.030** | **0.549 ± 0.029** | 1.764 ± 0.022 | 0.703 ± 0.007 | 1.721 ± 0.047 | 0.578 ± 0.014 | 1.658 ± 0.031 | 0.613 ± 0.031 | 1.823 ± 0.035 | 0.707 ± 0.074 |
| | 192 | **1.627 ± 0.042** | **0.672 ± 0.043** | 1.833 ± 0.032 | 0.873 ± 0.015 | 1.715 ± 0.047 | 0.691 ± 0.044 | 1.725 ± 0.031 | 0.759 ± 0.038 | 1.870 ± 0.019 | 0.844 ± 0.056 |
| | 336 | **1.658 ± 0.032** | **0.641 ± 0.048** | 1.838 ± 0.009 | 0.840 ± 0.011 | 1.763 ± 0.037 | 0.679 ± 0.035 | 1.765 ± 0.032 | 0.723 ± 0.031 | 1.870 ± 0.025 | 0.790 ± 0.059 |
| | 720 | **1.690 ± 0.029** | **0.646 ± 0.050** | 1.848 ± 0.023 | 0.815 ± 0.026 | 1.809 ± 0.013 | 0.805 ± 0.017 | 1.787 ± 0.012 | 0.710 ± 0.026 | 1.869 ± 0.031 | 0.766 ± 0.052 |

assigned to partitions while maintaining the same total number of partitions as the native subdataset configuration.

Table 9: Performance comparison of different granularity partitions.

| Dataset | Pred. length | Combine each two | | Native Subdataset | | Random Partition | |
|---|---|---|---|---|---|---|---|
| | | NLL | MAPE | NLL | MAPE | NLL | MAPE |
| ETTh1 | 96 | 1.837 ± 0.077 | 0.782 ± 0.029 | **1.760 ± 0.029** | **0.694 ± 0.020** | 1.965 ± 0.077 | 0.752 ± 0.029 |
| | 192 | 1.860 ± 0.032 | 0.740 ± 0.047 | **1.766 ± 0.022** | **0.682 ± 0.039** | 1.969 ± 0.028 | 0.723 ± 0.041 |
| | 336 | 1.885 ± 0.045 | 0.834 ± 0.045 | **1.825 ± 0.032** | **0.746 ± 0.038** | 1.998 ± 0.046 | 0.832 ± 0.033 |
| | 720 | 1.911 ± 0.058 | 0.991 ± 0.063 | **1.853 ± 0.025** | **0.947 ± 0.038** | 2.009 ± 0.054 | 0.934 ± 0.048 |

We observe two key findings that align with our assumptions. First, we assume that TSIS exhibits locality within partitions based on semantic similarity (such as time series domains); consequently, random partitioning performs worst due to the lack of semantic coherence. Second, coarser granularity make the locality within each partition weaker, thus the performance is worse than "Native Subdataset".

### D.4 SENSITIVITY TO DECAY FACTOR $\beta$

The performance in Table 10 is not very sensitive to the $\beta$ setting as long as they are set to a reasonable range, in our experiment throughout the paper, we choose $\beta = 0.01$.

Table 10: Performance comparison of different beta values.

| Dataset | Pred. length | $\beta$=0.1 | | $\beta$=0.01 | | $\beta$=0.001 | | $\beta$=0.0001 | |
|---|---|---|---|---|---|---|---|---|---|
| | | NLL | MAPE | NLL | MAPE | NLL | MAPE | NLL | MAPE |
| ETTh1 | 96 | $1.744 \pm 0.066$ | $0.814 \pm 0.033$ | $1.760 \pm 0.029$ | $\mathbf{0.694 \pm 0.020}$ | $\mathbf{1.688 \pm 0.066}$ | $0.758 \pm 0.049$ | $1.762 \pm 0.078$ | $0.739 \pm 0.039$ |
| | 192 | $1.729 \pm 0.041$ | $0.727 \pm 0.031$ | $1.766 \pm 0.022$ | $\mathbf{0.682 \pm 0.039}$ | $\mathbf{1.717 \pm 0.038}$ | $0.714 \pm 0.042$ | $1.754 \pm 0.038$ | $0.704 \pm 0.040$ |
| | 336 | $1.774 \pm 0.036$ | $0.835 \pm 0.022$ | $1.825 \pm 0.032$ | $\mathbf{0.746 \pm 0.038}$ | $\mathbf{1.753 \pm 0.035}$ | $0.783 \pm 0.023$ | $1.781 \pm 0.039$ | $0.787 \pm 0.023$ |
| | 720 | $1.807 \pm 0.054$ | $1.061 \pm 0.039$ | $1.853 \pm 0.025$ | $\mathbf{0.947 \pm 0.038}$ | $\mathbf{1.783 \pm 0.057}$ | $0.962 \pm 0.038$ | $1.793 \pm 0.056$ | $0.953 \pm 0.028$ |

## D.5 Sensitivity to SNR filter bar

We also conducted an experiment to examine how the SNR threshold affects performance, as shown in Table 11. The SNR threshold filtering is designed to remove noisy data points based on domain knowledge, complementing the influence score-based filtering. When the threshold is very strict (e.g., 5dB), a large number of data points are excluded by the hard filtering and degrade the result. An appropriate value of $k$ can be found at relatively small values. We acknowledge that more sophisticated methods for designing hyperparameters to filter noisy data points may exist, and we leave this exploration for future work. In our experiments throughout the paper, we use $k = 3$dB.

Table 11: Performance comparison of different SNR values.

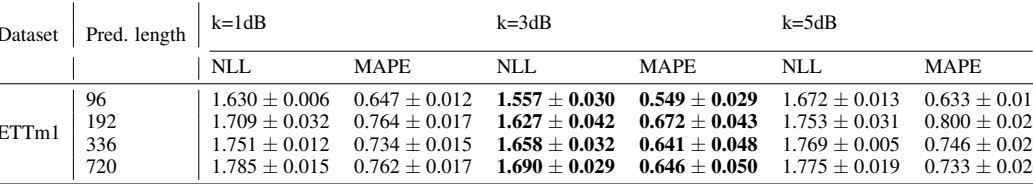

| Dataset | Pred. length | k=1dB | | k=3dB | | k=5dB | |
|---|---|---|---|---|---|---|---|
| | | NLL | MAPE | NLL | MAPE | NLL | MAPE |
| ETTm1 | 96 | $1.630 \pm 0.006$ | $0.647 \pm 0.012$ | $\mathbf{1.557 \pm 0.030}$ | $\mathbf{0.549 \pm 0.029}$ | $1.672 \pm 0.013$ | $0.633 \pm 0.013$ |
| | 192 | $1.709 \pm 0.032$ | $0.764 \pm 0.017$ | $\mathbf{1.627 \pm 0.042}$ | $\mathbf{0.672 \pm 0.043}$ | $1.753 \pm 0.031$ | $0.800 \pm 0.022$ |
| | 336 | $1.751 \pm 0.012$ | $0.734 \pm 0.015$ | $\mathbf{1.658 \pm 0.032}$ | $\mathbf{0.641 \pm 0.048}$ | $1.769 \pm 0.005$ | $0.746 \pm 0.022$ |
| | 720 | $1.785 \pm 0.015$ | $0.762 \pm 0.017$ | $\mathbf{1.690 \pm 0.029}$ | $\mathbf{0.646 \pm 0.050}$ | $1.775 \pm 0.019$ | $0.733 \pm 0.026$ |

## D.6 Sensitivity to validation set selection

Here we provide the result on a different number of validation sample sizes and the error bar in Table 12. The choice of data samples in the validation dataset is random and we calculate the error bar for sensitivity study. In our experiments throughout the paper, we use 32 as validation size.

Table 12: Performance comparison of different batch sizes.

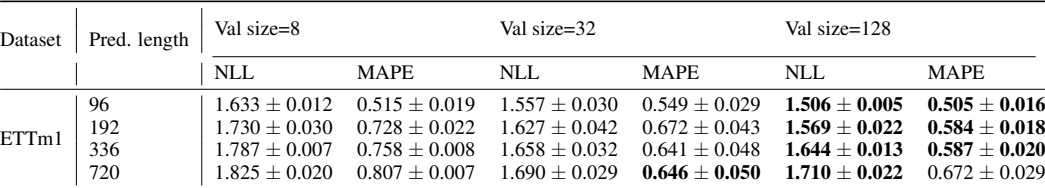

| Dataset | Pred. length | Val size=8 | | Val size=32 | | Val size=128 | |
|---|---|---|---|---|---|---|---|
| | | NLL | MAPE | NLL | MAPE | NLL | MAPE |
| ETTm1 | 96 | $1.633 \pm 0.012$ | $0.515 \pm 0.019$ | $1.557 \pm 0.030$ | $0.549 \pm 0.029$ | $\mathbf{1.506 \pm 0.005}$ | $\mathbf{0.505 \pm 0.016}$ |
| | 192 | $1.730 \pm 0.030$ | $0.728 \pm 0.022$ | $1.627 \pm 0.042$ | $0.672 \pm 0.043$ | $\mathbf{1.569 \pm 0.022}$ | $\mathbf{0.584 \pm 0.018}$ |
| | 336 | $1.787 \pm 0.007$ | $0.758 \pm 0.008$ | $1.658 \pm 0.032$ | $0.641 \pm 0.048$ | $\mathbf{1.644 \pm 0.013}$ | $\mathbf{0.587 \pm 0.020}$ |
| | 720 | $1.825 \pm 0.020$ | $0.807 \pm 0.007$ | $1.690 \pm 0.029$ | $\mathbf{0.646 \pm 0.050}$ | $\mathbf{1.710 \pm 0.022}$ | $0.672 \pm 0.029$ |

# E The Use of Large Language Models (LLMs)

We use LLM (ChatGPT) for language polishing to improve the grammar and the fluency. LLMs are **not** used for conceptual definitions, method design, data analysis.

