# OpenReview forum: "OATS: Online Data Augmentation for Time Series Foundation Models"
_ICLR.cc/2026/Conference — Submitted to ICLR 2026_

### Official Review · Reviewer_hNyo · 2025-10-26

**Soundness:** 3
**Presentation:** 1
**Contribution:** 3
**Rating:** 4
**Confidence:** 3

**Summary:**

This paper proposes a novel online data augmentation framework for Time Series Foundation Models (OATS). OATS dynamically generates new data at each training step by: (1) using a data attribution method (TSIS) to identify "high-quality" training samples based on their influence on a small validation set; (2) using these high-quality samples as conditional prompts for a diffusion model to generate new, synthetic data; and (3) employing an explore-exploit mechanism to manage the high computational cost. The experiments demonstrate that this dynamical approach outperforms regular training and static augmentation baselines.

**Strengths:**

1. **Novelty**: The core idea using a data attribution score for sampling, exploitation-exploration paradigm is very interesting and novel. The motivation, that data augmentation should be a dynamic process tailored to the model's state rather than a static, one-time operation, is well-motivated conceptual contribution.

2. **Principled Framework**: The paper proposes a complete, end-to-end framework. The use of data attribution (influence functions) to define data "quality" is a much more principled approach than the manual heuristics (jitter, mixup) it compares against.

3. **Efficiency Considerations**: The authors correctly identify the high computational cost of their method and proactively address it with a practical explore-exploit mechanism, including a sensible analysis of the cost-performance trade-off (Fig 4).

**Weaknesses:**

1. **No Standard Deviations**: Results in Tables 1 & 2 are reported from what appears to be a single run, with no random seeds mentioned. Without mean and std. dev. over multiple seeds, it is impossible to know if these gains are statistically significant or just random noise from a lucky run.

2. **Details on the Validation Set**: The entire method relies on calculating influence scores relative to a 32-sample validation set. The authors does not describe the composition of this set. This will affect the analysis a lot. For example, a high TSIS for the "solar power" subset might not mean it's "high quality" data; it might just mean the 32-sample validation set was heavily biased with solar power data.

3. **Poor Representation**: The paper presentation can be better. For example, line 244,
$\mathcal{B}_t \sim \pi_r^{\left|\mathcal{B}_t\right|}$​, which I believe this describes uniform batch sampling, but the notation is confusing. Also, Figure 1's numbering, the main figure goes from (1) $\rightarrow$ (3). I think step (2) and (3) are entangled. For example, Figure 4. No distinction between first row plots and second row, only caption mentions it.

4. **Unjustified Heuristics**: The TSIS score (Eq 1) combines a influence score with a completely unexplained, heuristic-based SNR hard filter $\left(I_{\mathrm{SNR}\left(z_i\right)<k} \cdot \infty\right)$. This point is unjustified.

**Questions:**

I am actually very positive about the proposed method, and am willing to raise the score to 6 or 8 as long the authors could justify my questions and resolve my concerns.

**Major concerns**
1. What's the random seeds you've used during the experiments and why did you not report any of the mean and std?

2. What's the exact composition of the validation set? Without this, I think the described analysis is less meaningful. For example, even if dataset portion is very low during training (e.x., PEMS07), if the validation dataset consists of huge amount of PEMS07, then the TSIS score for that data samples will be high.

3. Could you justify the inclusion of the hard-coded SNR filter in the TSIS calculation? What is the sensitivity to the threshold $k$, and why is this heuristic necessary on top of the influence score?

**Minor concerns**
1. $\infty$ is not a number, if you want to include $\infty$ value with SNR, I think you should change the range from $\mathbb{R}$ to $\mathbb{R} \cup \\{ \infty, -\infty \\}$.
2. Figure 4's y axis represents the different dataset, would it be possible to add dataset name on it? For example, NLL (dataset 1) or something..
3. Also, I think it would be nice if you could mention what 'overall' measurement does on Table 1, and 2 at experiments setup or somewhere that we can easily see, like evaluation metrics paragraph. It's average performance of using prediction lengths, but only mentioned in Table 1 caption.
4. You don't provide the proof of proposition 1 even though I think it's very obvious and just a few lines of proof. But as long as you mentioned it as proposition, I think you should provide the proof in Appendix.
5. Line 220, equation 3, would it be better for add 'a' and 'b' to 'm', so that change `+m` to `+am + b` to automatically control the bias? Would this improve the performance?

Again, I think the idea is cool and I am willing to raise the score as long as my concerns are justified or resolved.

---

> ### Author Response · Authors · 2025-11-22
>
> > Weakness 1 & Major concerns 1
>
> Thank you for pointing out the potential randomness in Table 1&2. We have included the error bar (standard error of mean over 5 runs) in updated Table 1&2 of the revised PDF.
>
>
> > Weakness 2 & Major concerns 2
>
> Thank you for pointing out the sensitivity of validation set selection. Here we provide the result on a different number of validation sample sizes and the error bar. The choice of data samples in the validation dataset is random and we calculate the error bar for sensitivity study. The result is shown in following table, more detailed experiment settings are included in the revised paper Appendix D.6:
>
> | Dataset | Pred. length | BS=8          |               | BS=32         |               | BS=128        |               |
> | ------- | ------------ | ------------- | ------------- | ------------- | ------------- | ------------- | ------------- |
> |         |              | **NLL**       | **MAPE**      | **NLL**       | **MAPE**      | **NLL**       | **MAPE**      |
> | ETTm1   | 96           | 1.633 ± 0.012 | 0.515 ± 0.019 | 1.557 ± 0.030 | 0.549 ± 0.029 | 1.506 ± 0.005 | 0.505 ± 0.016 |
> | ETTm1   | 192          | 1.730 ± 0.030 | 0.728 ± 0.022 | 1.627 ± 0.042 | 0.672 ± 0.043 | 1.569 ± 0.022 | 0.584 ± 0.018 |
> | ETTm1   | 336          | 1.787 ± 0.007 | 0.758 ± 0.008 | 1.658 ± 0.032 | 0.641 ± 0.048 | 1.644 ± 0.013 | 0.587 ± 0.020 |
> | ETTm1   | 720          | 1.825 ± 0.020 | 0.807 ± 0.007 | 1.690 ± 0.029 | 0.646 ± 0.050 | 1.710 ± 0.022 | 0.672 ± 0.029 |
>
> It shows that the larger the validation set, the better the performance. A small validation size (like 8) may degrade the performance because of potential data selection “overfitting”.
>
> > Weakness 3
>
> Thank you for the suggestion. We have made some revision to Figure 1, Figure 4 and line 244.
>
> > Weakness 4 & Major concerns 3
>
> We carried out an experiment studying how SNR threshold affects the result. The result is shown in following table, more detailed experiment settings are included in the revised paper Appendix D.5:
> | Dataset | Pred. length | k=1dB             |                   | k=3dB             |                   | k=5dB             |                   |
> |---------|--------------|-------------------|-------------------|-------------------|-------------------|-------------------|-------------------|
> |         |              | NLL               | MAPE              | NLL               | MAPE              | NLL               | MAPE              |
> | ETTm1   | 96           | 1.630 ± 0.006     | 0.647 ± 0.012     | 1.557 ± 0.030     | 0.549 ± 0.029     | 1.672 ± 0.013     | 0.633 ± 0.013     |
> |         | 192          | 1.709 ± 0.032     | 0.764 ± 0.017     | 1.627 ± 0.042     | 0.672 ± 0.043     | 1.753 ± 0.031     | 0.800 ± 0.022     |
> |         | 336          | 1.751 ± 0.012     | 0.734 ± 0.015     | 1.658 ± 0.032     | 0.641 ± 0.048     | 1.769 ± 0.005     | 0.746 ± 0.022     |
> |         | 720          | 1.785 ± 0.015     | 0.762 ± 0.017     | 1.690 ± 0.029     | 0.646 ± 0.050     | 1.775 ± 0.019     | 0.733 ± 0.026     |
>
> The intuition of introducing SNR threshold filtering is to remove some noisy data points based on domain knowledge, which work together with the influence score based filtering. If the filtering is very strict (e.g., 5dB), too many data points will be removed by the hard filtering so a proper k can be found at a relatively small value.
>
> > Minor concerns
>
> Thank you for the suggestion
>
> 1. We design (2) taking SNR as a hard filter, thus we use infinity as the factor. For later cached TSIS value, we emphasize it should be the “influence score part”.
> 2. We have updated the title to include the dataset name.
> 3. We update the evaluation metric paragraph and the caption of Table 1 and 2.
> 4. We update proposition 1 to include more details.
> 5. Thank you for the potential improvement direction, we may leave this to future work.

---

> > ### Comment · Reviewer_hNyo · 2025-11-26
> >
> > Thank you very much for the thoughtful response and the revisions.
> > After reviewing the updated manuscript and the new experiments, my main concerns have been addressed. I am happy to raise my score to 6.

---

> > > ### Author Response · Authors · 2025-11-26
> > >
> > > Thank you for your time and thoughtful feedback.

---

### Official Review · Reviewer_KZ1u · 2025-10-26

**Soundness:** 2
**Presentation:** 3
**Contribution:** 2
**Rating:** 4
**Confidence:** 4

**Summary:**

This paper proposes Online Data Augmentation for Time Series Foundation Models, a strategy for dynamically generating synthetic data during TSFM training. It uses data attribution scores TSIS, based on influence functions, to identify valuable training samples , which guide a conditional diffusion model for data generation. An explore-exploit mechanism aims to reduce computational cost. While addressing the relevant issue of adaptive augmentation, the method's reliance on approximations and heuristics raises concerns about its robustness and practicality.

**Strengths:**

1 Addresses the significant problem of improving TSFM training via adaptive augmentation, moving beyond static heuristics. The concept of integrating online data attribution with conditional generative modeling is an interesting synthesis.

2 Motivates the approach by highlighting limitations of existing methods. The overall framework is presented clearly. Using influence functions offers a principled motivation. Experiments show positive results compared to static baselines.

**Weaknesses:**

1 TSIS calculation relying on approximations (first-order Taylor, SGD assumption)  whose accuracy for large models with adaptive optimizers, like AdamW used here, is questionable and unverified. This undermines the reliability of the guiding signal.

2 The efficiency mechanism depends on the "locality of TSIS" heuristic, assuming similar influence within subsets. This may not hold generally, and performance could be sensitive to subset definition. Lack of sensitivity analysis for this and key hyperparameters (epsilon, beta) makes its robustness uncertain. And Crucial sensitivity analyses for SNR threshold k, and the validation set's impact are missing, hindering reproducibility and understanding the method's limits.

3 OATS adds significant complexity: conditional diffusion model & online influence calculation. The paper doesn't convincingly show that the performance gains justify this complexity, lacking comparing to potentially simpler adaptive methods.

4 TSIS calculation depends on a small validation set with 32 samples. Performance might be sensitive to this set's choice/quality, risking overfitting the augmentation strategy. This was not investigated.

**Questions:**

1 Could you provide stronger validation for the accuracy of the influence score approximation with large TSFMs and AdamW optimization? How reliable is the resulting guiding signal?

2 How robust is performance to the subset partitioning method? Can you provide evidence supporting the "locality of TSIS" assumption? Please provide sensitivity analyses for epsilon and beta & sensitivity analyses for key hyperparameters like the SNR threshold k and diffusion model parameters.

3 Can you provide a clearer cost-benefit analysis? How does OATS compare to simpler adaptive augmentation strategies? Do the gains justify the complexity across different scales/budgets?

4 Regarding the Validation Set D_val: Please provide analysis on how the choice, size (beyond 32 samples), and quality of D_val impact performance. How is overfitting to this set mitigated?

---

> ### Author Response · Authors · 2025-11-22
>
> Thank you for your insightful feedback. Here are our responses to your comments:
>
> > Weakness 1 & Question 1
>
> Thank you for pointing this out. We leverage first-order Taylor expansion on SGD optimization. A derivation specifically for Adam or AdamW is still an ongoing research direction for training data influence community. But some empirical results [1][2] have shown that leveraging SGD assumption and first-order Taylor expansion is good enough when the learning rate is small.
>
> [1] Wang, J. T., Song, D., Zou, J., Mittal, P., & Jia, R. (2024). Capturing the temporal dependence of training data influence. arXiv preprint arXiv:2412.09538.
>
> [2] Wang, J. T., Wu, T., Song, D., Mittal, P., & Jia, R. (2024). Greats: Online selection of high-quality data for llm training in every iteration. Advances in Neural Information Processing Systems, 37, 131197-131223.
>
> > Weakness 2 & Question 2
>
> Thank you for pointing out the sensitivity study of 1) training data partition, 2) decay factor, 3) epsilon and 4) SNR threshold k. I will discuss them one by one.
>
> ### Sensitivity study of training data partition
>
> In our experiments, we follow the native partition (the sub-dataset) contained in the LOTSA dataset. Here we carry out experiments on different settings of initial data granularities.  “Native Subdataset” is the setting we used in the paper; “Combine each two” means we use a coarser granularity by combining two native subdataset to one partition; Moreover, we also include a baseline by randomly splitting the data samples to different partitions (with the same number of native subdataset). Following is the result, more detailed experiment settings are included in the revised paper Appendix D.3:
>
> | Dataset | Pred. length | Combine each two  |                   | Native Subdataset |                   | Random Partition  |                   |
> |---------|--------------|-------------------|-------------------|-------------------|-------------------|-------------------|-------------------|
> |         |              | NLL               | MAPE              | NLL               | MAPE              | NLL               | MAPE              |
> | ETTh1   | 96           | 1.837 ± 0.077     | 0.782 ± 0.029     | 1.760 ± 0.029     | 0.694 ± 0.020     | 1.965 ± 0.077     | 0.752 ± 0.029     |
> |         | 192          | 1.860 ± 0.032     | 0.740 ± 0.047     | 1.766 ± 0.022     | 0.682 ± 0.039     | 1.969 ± 0.028     | 0.723 ± 0.041     |
> |         | 336          | 1.885 ± 0.045     | 0.834 ± 0.045     | 1.825 ± 0.032     | 0.746 ± 0.038     | 1.998 ± 0.046     | 0.832 ± 0.033     |
> |         | 720          | 1.911 ± 0.058     | 0.991 ± 0.063     | 1.853 ± 0.025     | 0.947 ± 0.038     | 2.009 ± 0.054     | 0.934 ± 0.048     |
>
> There are mainly two observations which follows our assumptions
>
> - We assume the locality of TSIS within a partition based on the semantic similarity (like time series domains), thus the random partition performs the worst.
>
> - Coarser granularity will harm the strength of locality, thus we observe a degradation when switching from Native Subdataset to Combine each two.

---

> ### Author Response · Authors · 2025-11-22
>
> > Weakness 2 & Question 2 (cont'd)
>
> ###  Decay factor
> We also perform sensitivity study to the decay factor β, The result is shown in following table, more detailed experiment settings are included in the revised paper Appendix D.4:
>
> | Dataset | Pred. length | β=0.1             |                   | β=0.01            |                   | β=0.001           |                   | β=0.0001          |                   |
> |---------|--------------|-------------------|-------------------|-------------------|-------------------|-------------------|-------------------|-------------------|-------------------|
> |         |              | NLL               | MAPE              | NLL               | MAPE              | NLL               | MAPE              | NLL               | MAPE              |
> | ETTh1   | 96           | 1.744 ± 0.066     | 0.814 ± 0.033     | 1.760 ± 0.029     | 0.694 ± 0.020     | 1.688 ± 0.066     | 0.758 ± 0.049     | 1.762 ± 0.078     | 0.739 ± 0.039     |
> |         | 192          | 1.729 ± 0.041     | 0.727 ± 0.031     | 1.766 ± 0.022     | 0.682 ± 0.039     | 1.717 ± 0.038     | 0.714 ± 0.042     | 1.754 ± 0.038     | 0.704 ± 0.040     |
> |         | 336          | 1.774 ± 0.036     | 0.835 ± 0.022     | 1.825 ± 0.032     | 0.746 ± 0.038     | 1.753 ± 0.035     | 0.783 ± 0.023     | 1.781 ± 0.039     | 0.787 ± 0.023     |
> |         | 720          | 1.807 ± 0.054     | 1.061 ± 0.039     | 1.853 ± 0.025     | 0.947 ± 0.038     | 1.783 ± 0.057     | 0.962 ± 0.038     | 1.793 ± 0.056     | 0.953 ± 0.028     |
>
> The performance is not very sensitive to the beta setting as long as they are set to a reasonable range, in our experiment throughout the paper, we choose β=0.01.
>
> ### Epsilon
> Figure 4 is the results of regular training and OATS with different \epsilon (0.3/0.5/0.7/1.0) on two validation datasets (ETTh1, Electricity) and four different prediction lengths.
>
> Though the best \epsilon is achieved on different values, most of the results follow a U-shaped pattern. This provides good empirical evidence that epsilon performs as a good handler of the explore-exploit trade-off. On the other hand, OATS with different \epsilon (0.3/0.5/0.7/1.0) significantly outperform regular training. The gap of evaluation metric between different \epsilon values is much smaller than that between regular training and OATS.
>
> ### SNR threshold k
> We carried out an experiment studying how SNR threshold affects the result. The result is shown in following table, more detailed experiment settings are included in the revised paper Appendix D.5:
>
> | Dataset | Pred. length | k=1dB             |                   | k=3dB             |                   | k=5dB             |                   |
> |---------|--------------|-------------------|-------------------|-------------------|-------------------|-------------------|-------------------|
> |         |              | NLL               | MAPE              | NLL               | MAPE              | NLL               | MAPE              |
> | ETTm1   | 96           | 1.630 ± 0.006     | 0.647 ± 0.012     | 1.557 ± 0.030     | 0.549 ± 0.029     | 1.672 ± 0.013     | 0.633 ± 0.013     |
> |         | 192          | 1.709 ± 0.032     | 0.764 ± 0.017     | 1.627 ± 0.042     | 0.672 ± 0.043     | 1.753 ± 0.031     | 0.800 ± 0.022     |
> |         | 336          | 1.751 ± 0.012     | 0.734 ± 0.015     | 1.658 ± 0.032     | 0.641 ± 0.048     | 1.769 ± 0.005     | 0.746 ± 0.022     |
> |         | 720          | 1.785 ± 0.015     | 0.762 ± 0.017     | 1.690 ± 0.029     | 0.646 ± 0.050     | 1.775 ± 0.019     | 0.733 ± 0.026     |
>
> The intuition of introducing SNR threshold filtering is to remove some noisy data points based on domain knowledge, which work together with the influence score based filtering. If the filtering is very strict (e.g., 5dB), too many data points will be removed by the hard filtering so a proper k can be found at a relatively small value.

---

> ### Author Response · Authors · 2025-11-22
>
> > Weakness 3 & Question 3
>
> We provide a detailed analysis of the scalability in Appendix B and we have updated the PDF to include more detailed information. In short, TSIS calculation will include at most 1X overhead in computation compared to regular training and on par for the memory footprint. The exploit-explore mechanism will reduce the computational cost by (1-\epsilon). The overhead ratio will not change when we have a larger TSFM or longer sequence length. The synthetic generation will be an additional overhead which is at most 0.5X overhead in computation compared to regular training. We also include empirical results at the same training time as regular training to show that OATS still performs the best.
>
> > Weakness 4 & Question 4
>
> Thank you for pointing out the sensitivity of validation set selection. Here we provide the result on a different number of validation sample sizes and the error bar. The choice of data samples in the validation dataset is random and we calculate the error bar for sensitivity study. The result is shown in following table, more detailed experiment settings are included in the revised paper Appendix D.6:
>
> | Dataset | Pred. length | BS=8          |               | BS=32         |               | BS=128        |               |
> | ------- | ------------ | ------------- | ------------- | ------------- | ------------- | ------------- | ------------- |
> |         |              | **NLL**       | **MAPE**      | **NLL**       | **MAPE**      | **NLL**       | **MAPE**      |
> | ETTm1   | 96           | 1.633 ± 0.012 | 0.515 ± 0.019 | 1.557 ± 0.030 | 0.549 ± 0.029 | 1.506 ± 0.005 | 0.505 ± 0.016 |
> | ETTm1   | 192          | 1.730 ± 0.030 | 0.728 ± 0.022 | 1.627 ± 0.042 | 0.672 ± 0.043 | 1.569 ± 0.022 | 0.584 ± 0.018 |
> | ETTm1   | 336          | 1.787 ± 0.007 | 0.758 ± 0.008 | 1.658 ± 0.032 | 0.641 ± 0.048 | 1.644 ± 0.013 | 0.587 ± 0.020 |
> | ETTm1   | 720          | 1.825 ± 0.020 | 0.807 ± 0.007 | 1.690 ± 0.029 | 0.646 ± 0.050 | 1.710 ± 0.022 | 0.672 ± 0.029 |
>
> It shows that the larger the validation set, the better the performance. A small validation size (like 8) may degrade the performance because of potential data selection “overfitting”.

---

> ### Comment · Reviewer_KZ1u · 2025-11-27
>
> I thank the authors for their detailed response and the significant effort put into the additional experiments during the rebuttal. I genuinely appreciate the innovation of the OATS framework and find the research direction very promising.
>
> However, I note that the crucial sensitivity analyses provided in the rebuttal e.g regarding SNR threshold, validation set size, and partitions were conducted primarily on the ETT datasets. This limitation raises potential concerns regarding the generalizability of the proposed heuristics to other domains. Given the complexity of the method, I believe it is essential to verify these sensitivity results on a wider variety of datasets with different characteristics and scales. Such comprehensive validation would significantly strengthen the paper and confirm the method's practical robustness.
>
> I will maintain my score of 4 at this stage. But I strongly encourage the authors to include these broader evaluations in future versions to make this a truly excellent research.

---

> > ### Author Response · Authors · 2025-12-03
> >
> > Thank you for your feedback and for recognizing the innovation of OATS. We're pleased that you find the research direction promising. Due to the time constraints of the rebuttal period, conducting large-scale experiments for every ablation study and sensitivity analysis is challenging. However, we will incorporate additional experiments in the final draft to address your concerns more comprehensively.

---

### Official Review · Reviewer_yySA · 2025-10-28

**Soundness:** 3
**Presentation:** 2
**Contribution:** 3
**Rating:** 4
**Confidence:** 4

**Summary:**

The paper introduces OATS, a dynamic data augmentation framework designed for large-scale time series foundation models. Unlike existing static augmentation methods, OATS continuously generates synthetic samples during training based on time-series influence scores that measure each sample’s contribution to model performance. High-quality samples, identified via TSIS, guide a diffusion-based conditional generation module to synthesize realistic time series. An explore–exploit mechanism balances computation and exploration of new data subsets. Empirical evaluations on six datasets and two TSFM architectures show consistent improvements in normalized MAPE and NLL, demonstrating that OATS outperforms TSMixup, Jitter, and regular training.

**Strengths:**

1. Novel online augmentation formulation. The paper introduces a principled online data augmentation framework tailored to time series foundation models, replacing heuristic static methods with influence-based selection. The originality and coherence of this formulation represent a meaningful advance in dataset optimization for TSFMs.

2. Explore–exploit mechanism for computational balance.
The explore–exploit design introduces a probabilistic scheduling approach that reuses cached influence scores while selectively updating them. This mechanism reduces redundant computation while maintaining adaptability to model dynamics.

**Weaknesses:**

1. Unclear definition and maintenance of cached subsets.
The paper assumes that the dataset is divided into L disjoint subsets and maintains exponentially moving averages of influence scores for each (Eq. 4), but the rationale for this partitioning is underexplained. There is limited discussion of how subset granularity or the decay factor β affects performance.

2. Limited baseline scope and fairness of comparison.
The experiments compare OATS only to Jitter and TSMixup, neglecting other adaptive or generative baselines. Additionally, the ratio of real to synthetic samples and the computational budget per method are not reported, which may affect fairness. The absence of statistical significance tests in Tables 1–2 weakens the claim of consistent superiority.

3. Incomplete details of the diffusion model configuration.
While Sec. 2.2 specifies the use of a conditional diffusion model, critical parameters such as the number of sampling steps, noise schedule, and conditioning dimensions are missing. The influence of class conditioning (Eq. 3) is not empirically isolated, and the qualitative results in Fig. 6 are descriptive but not quantified. As diffusion generation is a major computational component, more information is required to judge its efficiency and contribution.

**Questions:**

1. What evidence supports the claimed efficiency and scalability of OATS?
Can the authors report detailed GPU/CPU utilization, wall-clock time, and memory footprint for different ϵ values? How does the method perform on larger TSFMs or longer sequence lengths, and are there stability issues when scaling to these settings?

2. Could the diffusion model’s configuration and conditioning mechanism be detailed further?
What diffusion steps, noise variance schedules, and conditioning embedding sizes were adopted in Sec. 2.2? How much does the class-conditioning term in Eq. 3 improve results compared to an unconditional diffusion model?

3. Can the authors provide a quantitative justification for the TSIS approximation?
Would it be possible to include either an empirical correlation between first-order influence estimates and actual validation loss changes, or a theoretical error bound derived from Proposition 1? How sensitive is TSIS to gradient noise or varying learning rates during training?

---

> ### Author Response · Authors · 2025-11-22
>
> Thank you for your insightful feedback. Here are our responses to your comments:
>
> > Weakness 1
>
> Thank you for pointing out the clarification of the dataset partition and the sensitivity of the model performance with various designs of granularity and decay factor. In the explore-exploit mechanism, we assume locality of TSIS within each partition (because of semantic similarity) and along the training process. We will leverage the following two additional experiments to verify such an assumption.
>
> In our experiments, we follow the native partition (the sub-dataset) contained in the LOTSA dataset. Here we carry out experiments on different settings of initial data granularities.  “Native Subdataset” is the setting we used in the paper; “Combine each two” means we use a coarser granularity by combining two native subdataset to one partition; Moreover, we also include a baseline by randomly splitting the data samples to different partitions (with the same number of native subdataset). Following is the result, more detailed experiment settings are included in the revised paper Appendix D.3:
>
> | Dataset | Pred. length | Combine each two  |                   | Native Subdataset |                   | Random Partition  |                   |
> |---------|--------------|-------------------|-------------------|-------------------|-------------------|-------------------|-------------------|
> |         |              | NLL               | MAPE              | NLL               | MAPE              | NLL               | MAPE              |
> | ETTh1   | 96           | 1.837 ± 0.077     | 0.782 ± 0.029     | 1.760 ± 0.029     | 0.694 ± 0.020     | 1.965 ± 0.077     | 0.752 ± 0.029     |
> |         | 192          | 1.860 ± 0.032     | 0.740 ± 0.047     | 1.766 ± 0.022     | 0.682 ± 0.039     | 1.969 ± 0.028     | 0.723 ± 0.041     |
> |         | 336          | 1.885 ± 0.045     | 0.834 ± 0.045     | 1.825 ± 0.032     | 0.746 ± 0.038     | 1.998 ± 0.046     | 0.832 ± 0.033     |
> |         | 720          | 1.911 ± 0.058     | 0.991 ± 0.063     | 1.853 ± 0.025     | 0.947 ± 0.038     | 2.009 ± 0.054     | 0.934 ± 0.048     |
>
> There are mainly two observations which follows our assumptions
>
> - We assume the locality of TSIS within a partition based on the semantic similarity (like time series domains), thus the random partition performs the worst.
>
> - Coarser granularity will harm the strength of locality, thus we observe a degradation when switching from Native Subdataset to Combine each two.
>
> We also perform sensitivity study to the decay factor β, The result is shown in following table, more detailed experiment settings are included in the revised paper Appendix D.4:
>
> | Dataset | Pred. length | β=0.1             |                   | β=0.01            |                   | β=0.001           |                   | β=0.0001          |                   |
> |---------|--------------|-------------------|-------------------|-------------------|-------------------|-------------------|-------------------|-------------------|-------------------|
> |         |              | NLL               | MAPE              | NLL               | MAPE              | NLL               | MAPE              | NLL               | MAPE              |
> | ETTh1   | 96           | 1.744 ± 0.066     | 0.814 ± 0.033     | 1.760 ± 0.029     | 0.694 ± 0.020     | 1.688 ± 0.066     | 0.758 ± 0.049     | 1.762 ± 0.078     | 0.739 ± 0.039     |
> |         | 192          | 1.729 ± 0.041     | 0.727 ± 0.031     | 1.766 ± 0.022     | 0.682 ± 0.039     | 1.717 ± 0.038     | 0.714 ± 0.042     | 1.754 ± 0.038     | 0.704 ± 0.040     |
> |         | 336          | 1.774 ± 0.036     | 0.835 ± 0.022     | 1.825 ± 0.032     | 0.746 ± 0.038     | 1.753 ± 0.035     | 0.783 ± 0.023     | 1.781 ± 0.039     | 0.787 ± 0.023     |
> |         | 720          | 1.807 ± 0.054     | 1.061 ± 0.039     | 1.853 ± 0.025     | 0.947 ± 0.038     | 1.783 ± 0.057     | 0.962 ± 0.038     | 1.793 ± 0.056     | 0.953 ± 0.028     |
>
> The performance is not very sensitive to the beta setting as long as they are set to a reasonable range, in our experiment throughout the paper, we choose β=0.01.

---

> ### Author Response · Authors · 2025-11-22
>
> > Weakness 2
>
> Thank you for the suggestion for adding more baselines.
>
> We add a baseline using data-driven generative models (unconditional diffusion model, termed as “DD”) to generate synthetic data in an offline paradigm. The results show that our method could significantly outperform all baselines. The following table shows the result; more detailed experiment settings are included in the revised paper, Appendix D.2:
>
> | Dataset     | Pred. length | OATS          |               | DD            |               | Jitter        |               | MixUp         |               | Regular       |               |
> | ----------- | ------------ | ------------- | ------------- | ------------- | ------------- | ------------- | ------------- | ------------- | ------------- | ------------- | ------------- |
> |             |              | **NLL**       | **MAPE**      | **NLL**       | **MAPE**      | **NLL**       | **MAPE**      | **NLL**       | **MAPE**      | **NLL**       | **MAPE**      |
> | ETTm1       | 96           | 1.557 ± 0.030 | 0.549 ± 0.029 | 1.764 ± 0.022 | 0.703 ± 0.007 | 1.721 ± 0.047 | 0.578 ± 0.014 | 1.658 ± 0.031 | 0.613 ± 0.031 | 1.823 ± 0.035 | 0.707 ± 0.074 |
> | ETTm1       | 192          | 1.627 ± 0.042 | 0.672 ± 0.043 | 1.833 ± 0.032 | 0.873 ± 0.015 | 1.715 ± 0.047 | 0.691 ± 0.044 | 1.725 ± 0.031 | 0.759 ± 0.038 | 1.870 ± 0.019 | 0.844 ± 0.056 |
> | ETTm1       | 336          | 1.658 ± 0.032 | 0.641 ± 0.048 | 1.838 ± 0.009 | 0.840 ± 0.011 | 1.763 ± 0.037 | 0.679 ± 0.035 | 1.765 ± 0.032 | 0.723 ± 0.031 | 1.870 ± 0.025 | 0.790 ± 0.059 |
> | ETTm1       | 720          | 1.690 ± 0.029 | 0.646 ± 0.050 | 1.848 ± 0.023 | 0.815 ± 0.026 | 1.809 ± 0.013 | 0.805 ± 0.017 | 1.787 ± 0.012 | 0.710 ± 0.026 | 1.869 ± 0.031 | 0.766 ± 0.052 |
>
> The ratio of real to synthetic samples is set to 1:1 for OATS and all other baselines; we also include a comprehensive computation complexity analysis in Appendix B. The error bars (standard error of mean over 5 runs) are also included in the table 1 and 2 of the updated PDF.
>
> > Weakness 3 & Question 2
>
>
> Thank you for pointing this out the effectiveness and clarification of the diffusion model for synthetic data generation. We included results in Appendix D.1 to compare the generation from diffusion model with and without the class embedding.
>
> We have also included a detailed experiment setting for the generation model in Appendix C. In detail, the sampling steps of training is 200, noise schedule is a linear noise schedule ranging from 0.0005 to 0.1 and class conditioning dimension is 64.
>
> > Question 1
>
> We provide a detailed analysis of the scalability in Appendix B and we have updated the PDF to include more detailed information. In short, TSIS calculation will involve at most 1X overhead in computation compared to regular training and on par for the memory footprint. The exploit-explore mechanism will reduce the computational cost by (1-\epsilon). The overhead ratio will not change when we have a larger TSFM or longer sequence length. The synthetic generation will be an additional overhead which is at most 0.5X overhead in computation compared to regular training. We also include empirical results at the same training time as regular training to show that OATS still performs the best.
>
> > Question 3
>
> Thank you for pointing out the error of TSIS approximation. As stated in Proposition 1, the error is at the level of O(eta^2), which is substantially small when the learning rate is small.

---

### Official Review · Reviewer_ges9 · 2025-10-31

**Soundness:** 3
**Presentation:** 3
**Contribution:** 3
**Rating:** 4
**Confidence:** 3

**Summary:**

The paper introduces OATS, an online data augmentation method for pre-training time series foundation models on data across diverse domains. OATS estimates per-sample value from training dynamics, trains a diffusion-based framework conditioned on those high-value signals, and adopts an explore-exploit mechanism to balance quality and diversity. Experiments on multiple datasets show that OATS outperforms standard augmentation baselines.

**Strengths:**

1. Dynamically augmenting time series according to evolving training signals is important and well motivated.

2. The proposed method of sample selection, a diffusion generator, and an explicit explore–exploit mechanism is new in this context.

3. Experiments on multiple time series datasets demonstrate the effectiveness of the proposed method over standard augmentation methods.

**Weaknesses:**

1. The compared data augmentation methods, such as mix-up augmentations and jittering, are basic and limited. Another type of time series augmentation methods is based on training generative models [1]. Additionally, there exists work that [2] proposes similar online data augmentation by selecting high-quality data.

2. Evaluation of time series foundation models is limited to 6 datasets. Consider adding more diverse, standardized time series foundation model benchmarks such as GIFT-EVAL to better demonstrate the method's performance. Moreover, how does the proposed method compare with Chronos and Moirai on these benchmarks?

3. Results may depend on the initial data partitions/subsets. How sensitive are the final results to initial partitions?

4. Performance appears sensitive to the explore–exploit ratio and is not consistent across prediction lengths and datasets.

[1] Time Series Data Augmentation for Deep Learning: A Survey

[2] Filter, Augment, Forecast: Online Data Selection for Robust Time Series Forecasting

**Questions:**

1. Could you explain more on time series prototypes and how many prototypes are generated?

2. For datasets where OATS increases the proportion of selected/augmented samples during training, what patterns did you observe? Do these correspond to “harder” datasets?

---

> ### Author Response · Authors · 2025-11-22
>
> Thank you for your insightful feedback. Here are our responses to your comments:
>
> > Weakness 1
>
> Thank you for the suggestion for adding more baselines. We add a baseline using data-driven generative models (unconditional diffusion model, termed as “DD”) to generate synthetic data in an offline paradigm. The results show that our method could significantly outperform all baselines. The following table shows the result; more detailed experiment settings are included in the revised paper, Appendix D.2:
>
> | Dataset     | Pred. length | OATS          |               | DD            |               | Jitter        |               | MixUp         |               | Regular       |               |
> | ----------- | ------------ | ------------- | ------------- | ------------- | ------------- | ------------- | ------------- | ------------- | ------------- | ------------- | ------------- |
> |             |              | **NLL**       | **MAPE**      | **NLL**       | **MAPE**      | **NLL**       | **MAPE**      | **NLL**       | **MAPE**      | **NLL**       | **MAPE**      |
> | ETTm1       | 96           | 1.557 ± 0.030 | 0.549 ± 0.029 | 1.764 ± 0.022 | 0.703 ± 0.007 | 1.721 ± 0.047 | 0.578 ± 0.014 | 1.658 ± 0.031 | 0.613 ± 0.031 | 1.823 ± 0.035 | 0.707 ± 0.074 |
> | ETTm1       | 192          | 1.627 ± 0.042 | 0.672 ± 0.043 | 1.833 ± 0.032 | 0.873 ± 0.015 | 1.715 ± 0.047 | 0.691 ± 0.044 | 1.725 ± 0.031 | 0.759 ± 0.038 | 1.870 ± 0.019 | 0.844 ± 0.056 |
> | ETTm1       | 336          | 1.658 ± 0.032 | 0.641 ± 0.048 | 1.838 ± 0.009 | 0.840 ± 0.011 | 1.763 ± 0.037 | 0.679 ± 0.035 | 1.765 ± 0.032 | 0.723 ± 0.031 | 1.870 ± 0.025 | 0.790 ± 0.059 |
> | ETTm1       | 720          | 1.690 ± 0.029 | 0.646 ± 0.050 | 1.848 ± 0.023 | 0.815 ± 0.026 | 1.809 ± 0.013 | 0.805 ± 0.017 | 1.787 ± 0.012 | 0.710 ± 0.026 | 1.869 ± 0.031 | 0.766 ± 0.052 |
>
> Thanks for pointing out [1] proposes an online data augmentation algorithm that leverages an RHO-based method to perform data selection followed by traditional time series transformations such as jittering, which is closely related to our work. However, the simplicity of their synthetic data generation method may result in undesirable performance degradation. Moreover, the RHO-based method requires training a reference model, which can be computationally expensive. We have included discussion in the related work section in the revised PDF. The lack of official implementation makes it hard to reproduce their algorithm in a short time. We will include a comparison if enough implementation details are released or replied to by the authors of [1].
>
> [1] Taga, E. O., Gozeten, H. A., Tire, K., Dalvi, R., Heckel, R., & Oymak, S. Filter, Augment, Forecast: Online Data Selection for Robust Time Series Forecasting. In 1st ICML Workshop on Foundation Models for Structured Data.
>
> > Weakness 2
>
> Thank you for pointing out the validation set coverage. The proposed online data augmentation method is compared under the same data and model setting with regular training and other data augmentation baselines. We follow the same model (both encoder-only and decoder-only) settings as in [2] to keep a fair comparison. We selected the 6 test datasets to cover the full LSF dataset used in [2], and they are widely used to demonstrate the model's performance. We will include more standardized time series foundation model benchmarks like GIFT-EVAL in the future.
>
> [2] Yao, Q., Yang, C. H. H., Jiang, R., Liang, Y., Jin, M., & Pan, S. (2024). Towards neural scaling laws for time series foundation models. arXiv preprint arXiv:2410.12360.

---

> ### Author Response · Authors · 2025-11-22
>
> > Weakness 3
>
> Thank you for pointing out the factor of initial data partitions/subsets. In our experiments, we follow the native partition (the sub-dataset) contained in the LOTSA dataset. Here we carry out experiments on different settings of initial data granularities.  “Native Subdataset” is the setting we used in the paper; “Combine each two” means we use a coarser granularity by combining two native subdataset to one partition; Moreover, we also include a baseline by randomly splitting the data samples to different partitions (with the same number of native subdataset). Following is the result, more detailed experiment settings are included in the revised paper Appendix D.3:
>
> | Dataset | Pred. length | Combine each two  |                   | Native Subdataset |                   | Random Partition  |                   |
> |---------|--------------|-------------------|-------------------|-------------------|-------------------|-------------------|-------------------|
> |         |              | NLL               | MAPE              | NLL               | MAPE              | NLL               | MAPE              |
> | ETTh1   | 96           | 1.837 ± 0.077     | 0.782 ± 0.029     | 1.760 ± 0.029     | 0.694 ± 0.020     | 1.965 ± 0.077     | 0.752 ± 0.029     |
> |         | 192          | 1.860 ± 0.032     | 0.740 ± 0.047     | 1.766 ± 0.022     | 0.682 ± 0.039     | 1.969 ± 0.028     | 0.723 ± 0.041     |
> |         | 336          | 1.885 ± 0.045     | 0.834 ± 0.045     | 1.825 ± 0.032     | 0.746 ± 0.038     | 1.998 ± 0.046     | 0.832 ± 0.033     |
> |         | 720          | 1.911 ± 0.058     | 0.991 ± 0.063     | 1.853 ± 0.025     | 0.947 ± 0.038     | 2.009 ± 0.054     | 0.934 ± 0.048     |
>
> There are mainly two observations which follows our assumptions
>
> - We assume the locality of TSIS within a partition based on the semantic similarity (like time series domains), thus the random partition performs the worst.
>
> - Coarser granularity will harm the strength of locality, thus we observe a degradation when switching from Native Subdataset to Combine each two.
>
> > Weakness 4
>
> Figure 4 shows the results of regular training and OATS with different \epsilon (0.3/0.5/0.7/1.0) on two validation datasets (ETTh1, Electricity) and four different prediction lengths.
>
> Though the best \epsilon is achieved on different values, most of the results follow a U-shaped pattern. This provides good empirical evidence that epsilon performs as a good handler of the explore-exploit trade-off. On the other hand, OATS with different \epsilon (0.3/0.5/0.7/1.0) significantly outperform regular training. The gap of evaluation metric between different \epsilon values is much smaller than that between regular training and OATS.

---

> ### Author Response · Authors · 2025-11-22
>
> > Question 1
>
> The time series prototype module is designed to help with high-quality data-guided time series generation, which can extract subset-specific weight for a specific data sample and use it combined with class embedding as the generation condition. The prototype can be seen as predefined “words,” and different data samples may have different weights stimulated by these prototypes. Here we use a fixed number of prototypes to be 16.
>
> > Question 2
>
> Thank you for pointing out this. We have two observations during the case study: 1) The preference according to TSIS changes rapidly in the first stage (first few epochs) while becoming more stable later. 2) Some training subdataset (e.g., solar) is preferred by several validation sets. We take it as a future study to further improve the understanding of time series quality.

---

### Meta-Review · Area_Chair_EJ9s · 2026-01-04

**Summary:**

The paper proposes OATS, a framework that dynamically generates synthetic training data during time series foundation model training. The core idea is to use data attribution scores to identify high-quality training samples, which then guide a conditional diffusion model to generate synthetic data tailored to each training step. An explore-exploit mechanism balances computational efficiency with adaptation to evolving training dynamics. Experiments on six datasets and two TSFM architectures show consistent improvements over static augmentation baselines and regular training.

**Reviewer Concerns:**

Reviewers raised several concerns. Some were addressed in the response, including: missing error bars/statistical significance, lack of generative model baselines, and missing sensitivity analyses for hyperparameters.

However, other concerns remain unresolved:
(1) sensitivity analyses were conducted primarily on ETT datasets;
(2) the validity of first-order Taylor approximation for TSIS with AdamW optimization relies on external citations rather than direct validation; (3) evaluation scope is limited to six datasets without comparison to established TSFMs or standardized benchmarks.

Additionally, the AC identified the following concerns:

(4) the related work section omits directly relevant prior art on adaptive/learned augmentation, including MODALS (ICLR 2021), AutoTCL (ICLR 2024), AutoAugment (CVPR 2019), etc.
(5) evaluation metrics (both NLL and MAPE) exhibit non-monotonic behavior across prediction lengths (e.g., NLL decreases from 96→336 in ETT datasets), which contradicts expected forecasting behavior and is neither acknowledged nor explained;
(6) no justification is provided for highlighting prediction length 192 in main tables over other horizons.
(7) typos.   For example:
a) usage of full name of abbreviations are used arbitrarily.
b) additional dot at 2.4 algorithm .
c) b/2 should be⌊b/2⌋
d) lack of comma eq 4.

**Reviewer Scores:**

4,4,4,4 -> 4,4,4,6

One reviewer explicitly raised their score to 6 after the rebuttal, satisfied with the addition of error bars, SNR sensitivity analysis, and validation set size experiments. This is a confirmed post-rebuttal score.

---

### Decision · Program_Chairs · 2026-01-26

Reject